# Delving into Sequential Patches for Deepfake Detection

Jiazhi Guan[1,2], Hang Zhou[2], Zhibin Hong[2], Errui Ding[2],
Jingdong Wang[2], Chengbin Quan[1], Youjian Zhao[1,3*]

[1]Department of Computer Science and Technology, Tsinghua University
[2]Department of Computer Vision Technology (VIS), Baidu Inc.   [3]Zhongguancun Laboratory
{guanjz20@mails., quancb@, zhaoyoujian@}tsinghua.edu.cn
{zhouhang09, dingerrui, wangjingdong}@baidu.com

## Abstract

Recent advances in face forgery techniques produce nearly visually untraceable deepfake videos, which could be leveraged with malicious intentions. As a result, researchers have been devoted to deepfake detection. Previous studies have identified the importance of local low-level cues and temporal information in pursuit to generalize well across deepfake methods, however, they still suffer from robustness problem against post-processings. In this work, we propose the **L**ocal- & **T**emporal-aware **T**ransformer-based Deepfake **D**etection (**LTTD**) framework, which adopts a local-to-global learning protocol with a particular focus on the valuable temporal information within local sequences. Specifically, we propose a Local Sequence Transformer (LST), which models the temporal consistency on sequences of restricted spatial regions, where low-level information is hierarchically enhanced with shallow layers of learned 3D filters. Based on the local temporal embeddings, we then achieve the final classification in a global contrastive way. Extensive experiments on popular datasets validate that our approach effectively spots local forgery cues and achieves state-of-the-art performance.

## 1 Introduction

With the development of face forgery methods [1, 25, 26, 66, 28, 23, 37], an enormous amount of fake videos (a.k.a deepfakes) have raised non-neglectable concerns on privacy preservation and information security. To this end, researches have been devoted to the reliable tagging of deepfakes in order to block the propagation of malicious information. However, it is still an open problem due to the limited generalization of detection methods and the continuous advances in deepfake creation.

Earlier studies [7, 34, 4, 13, 14, 63, 45, 69] devote efforts to enhance general Convolutional Neural Networks (CNN) for identifying clear semantic distortions in deepfakes. Recent methods pay attention to the temporal inconsistency problem [6, 33, 21, 58, 51, 60, 50, 5], yet most of them fall into semantic motion understanding (e. g., detecting abnormal eye blinking, phoneme-viseme mismatches, aberrant landmark fluctuation). These methods are able to learn specific forgery patterns, however, the remarkable visual forgery cues are expected to be gradually eliminated during the continuous arms race between forgers and detectors. As a result, the *generalizability* of previous methods is typically unsatisfactory when encountering deepfakes generated by unseen techniques.

On the other hand, the importance of low-level information is identified for tackling the *generalization* problem. A group of studies are developed using hand-made low-level filters (e. g., DCT [44, 19, 29], steganalysis features [67], SRM [40]) to better capture subtle differences between generated textures and the natural ones. However, methods depending on recognizable low-level patterns would become less effective on degraded data with commonly applied post-processing procedures like visual compression [39, 21, 65], which indicates the lack of *robustness*.

---

[*]Corresponding author.

In this work, we rethink the appropriate representation that can ensure both *generalizability* and *robustness* in deepfake detection. The inspiration is taken from Liu et al. [39], which indicates that deepfakes can be directly distinguished from pieces of skin patches. Such practice prevents network from overfitting to global semantic cues, making learned patterns more *generalizable*. In addition, as the creation of deepfakes inevitably relies on frame re-assembling, the substantial temporal differences also arise locally during the *independent local modifications* of forged frames. As the underlying temporal patterns are less affected by spatial interference, such temporal inconsistency will not be easily erased during common perturbations, making the low-level modeling more *robust*.

Motivated by the observations above, we propose the **L**ocal- & **T**emporal-aware **T**ransformer-based Deepfake **D**etection (**LTTD**) framework, which particularly focuses on patch sequence modeling in deepfake detection with Transformers [17]. Detailedly, we divide the 3D video information spatially into independent local regions (as shown in Fig. 1). Encouraged by the recent success of vision transformers [17], we formulate the patch sequence modeling problem in a self-attention style and propose a *Local Sequence Transformer (LST)* module, which operates on sequences of local patches. Benefited from the attention mechanism, LST is not constrained by the receptive field in CNNs, allowing for better learning of both long and short span temporal patterns. In addition, we hierarchically inject low-level results from shallow 3D convolutions after self-attention layers to enhance the low-level feature learning at multiple scales. Such design particularly emphasizes on the low-level information from a temporal consistency perspective.

After modeling each sequence of local patches in LST, two questions remain to be solved: 1) how to model their inherent relationships and 2) how to aggregate their information for final prediction. We explicitly impose global contrastive supervision on patch embeddings with a *Cross-Patch Inconsistency (CPI)* loss. Then the final predictions is given in the *Cross-Patch Aggregation (CPA)* module with follow up Transformer blocks. Ablations on these designs show non-trivial improvements.

Our contributions can be summarized as follows: **1)** We propose the Local- & Temporal-aware Transformer-based Deepfake Detection (LTTD) framework, which emphasizes low-level local temporal inconsistency by modeling sequences of local patches with Transformer. **2)** We design the Cross-Patch Inconsistency (CPI) loss and the Cross-Patch Aggregation (CPA) module, which efficiently aggregate local information for global prediction. **3)** Quantitative experiments show that our approach achieves the state-of-the-art generalizability and robustness. Qualitative results further illustrate its interpretability.

## 2 Related work

### 2.1 Deepfake detection

In recent years, we witness great progress in deepfake detection, where numerous forgery spotting models are proposed successively to address the practical demands of the application. In the earlier stage, methods [7, 34, 4, 14, 45, 69] are built with a major emphasis on spotting semantic visual artifacts with sophisticated model designs. Dang et al. [13] design a segmentation task optimized with the classification backbone simultaneously to predict the forgery regions. Zhao et al. [63] regard the deepfake detection task as a fine-grained classification problem. While these methods achieve satisfied in-dataset results, the cross-dataset evaluation shows poor generalizability.

Nevertheless, generalizability is considered to be the first point should it be designed for practical application scenarios, since we never have a chance to foresee the attackers' movements. More works [56, 10, 49] begin to notice the vital problem. Wang et al. [56] argue that the lack of generalizability is due to overfitting to significant semantic visual artifacts, and propose a dynamic data argumentation schema to relieve the issue. But considering the CNN always takes the downsampled semantic representation for final classification, their method tends to only focus on more semantic visual artifacts. Another group of works [31, 10, 64, 44, 19, 29, 40, 36, 39, 9, 20] dig deeper into the fundamental differences of deepfakes from the generation process. With the expectation that forgery methods will gradually improve, those works propose to identify deepfakes from low-level image features rather than semantic visual clues, as the latter are disappearing in the latest deepfakes. [31, 10, 64] both notice the content-independent low-level features that can uniquely identify their sources and the identity swapping will destroy the origin consistency. Li et al. [31] propose to detect those low-level features across facial boundary. Zhao et al. [64] and Chen et al. [10] turn to learn the spatial-local inconsistencies. Other methods look for clues in the frequency domain. With

DCT transform, [44, 19, 29] fuse the low-level frequency pattern learning into CNN to improve the generalizability. Liu et al. [36] instead theoretically analyze that phase information are more sensitive to upsampling, therefore, such low-level feature is more crucial than high-level semantic information for our task. The finds of [39] further encourage us to detect deepfakes from local patches, where the generated images are easy to be isolated even only one small skin patch is given. However, there is still a vital issue that low-level features should be even less robust to real-world distortions. Experiments in [39] show that simple smoothing could impair the performance of more than 20%. Drastic performance drops of [31] on suspected videos with degradation also indicate the weakness of low-level pattern learning. Different from current works, we propose to rely on temporal low-level changes with learnable shallow filters to better cope with real-world degradations.

Although temporal methods [6, 33, 21, 58, 50, 65] are also studied, most of they fall into the visual anomalous pattern learning such as abnormal eye blinking [33], non-synchronized lip movements [6, 21], inconsistent facial movements [65], etc. In the foreseeable further or even good cases of current arts, we could hardly find these significant patterns. In contrast, we explore the substantial temporal inconsistency of independently frame-wise generation at local patches, which is more generalizable and also more robust to common perturbations.

### 2.2 Vision transformer

Vaswani et al. [55] first propose to using only self-attention, multilayer perceptron, and layer norm to establish a new canonical form, coined Transformer, for natural language processing (NLP). Promoted by the great advancement in NLP, researchers of vision communities also start to explore the potential of transformer designs. ViT [17] could be the first success to apply pure typical transformer in image classification. After that, many researches also extend transformer into different vision tasks, (e. g., semantic segmentation [18], action recognition [43, 32], video understanding [38, 8, 62], scene graph generation [12], object detection [68, 59]) and achieve exceptional performance.

Recently, transformer structure is also adopted by a few works [65, 16, 57] in deepfake detection. [57] stack multiple ViTs [17] with different spatial embedding size. With the motivation to learn identity information, a transformer with an extra id-token is proposed in [16]. Besides, the closest work to us, [65] adopt few self-attention layers to aggregate successive frames' features, which are completely downsampled semantic embeddings. While in our method, transformer is introduced to achieve patch-sequential temporal learning in a restricted spatial receptive field with a totally different purpose to identify low-level temporal inconsistency.

## 3 Approach

In this section, we will elaborate the proposed **L**ocal- & **T**emporal-aware **T**ransformer-based Deepfake **D**etection (**LTTD**) framework after formally define the problem in section 3.1. We describe Local Sequence Transformer (**LST**) in section 3.2. Then, the Cross-Patch Inconsistency (**CPI**) loss is introduced with the Cross-Patch Aggregation (**CPA**) module in section 3.3.

### 3.1 Problem statement

Given one image $\mathbf{x} \in \mathbb{R}^{C \times H \times W}$, we reshape it into a sequence of flattened 2D patches $\{\mathbf{x}^i \in \mathbb{R}^{C \cdot P^2} | i = 1, 2, ..., N\}$, where $C$ is the number of image channel, $P$ is the patch size, and $N = HW/P^2$ is the patch number. In the original setting of ViT [17], the separated patches can be directly sent into the Transformer blocks for semantic understanding. Differently, in our task, we take a video clip $\mathbf{v} \in \mathbb{R}^{T \times C \times H \times W}$ as input, where the extra $T$ indicates the clip length, i. e., with $T$ successive frames. We also split each frame into independent patches for better low-level forgery pattern learning, but with an additional temporal dimension. The flattened patch set is represented as $\mathbf{s} = \{\mathbf{x}^{t,i} \in \mathbb{R}^{C \cdot P^2} | t = 1, 2, ..., T; i = 1, 2, ..., N\}$, where $\mathbf{x}^{t,i}$ stands for the flattened patch at the $i$-th spatial region of the $t$-th frame. Thus the total patch number becomes $T \cdot N$. After that, our proposed LTTD leverages the patch locality with learnable filters and powerful self-attention operations to explore the low-level temporal inconsistency of deepfakes, and finally give a prediction (fake or real) based on global contrast across the whole spatial region.

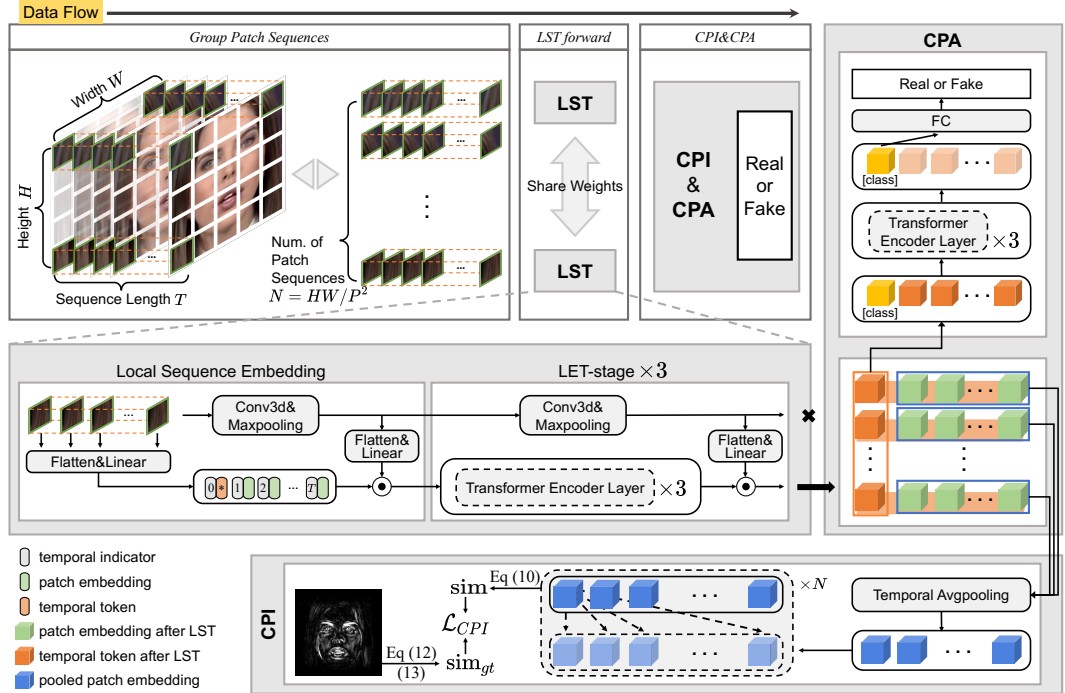

Figure 1: The overall pipeline and details of the proposed **L**ocal- & **T**emporal-aware **T**ransformer-based Deepfake **D**etection (**LTTD**). **Top left**: we divide the whole process into three cascaded parts as *Group Patch Sequences*, *LST forward*, and *CPI&CPA*. **Others**: we illustrate the details of Local Sequence Transformer (LST), Cross-Patch Inconsistency (CPI), and Cross-Patch Aggregation (CPA).

### 3.2 Local Sequence Transformer

As we discussed previously, in order to learn the low-level temporal patterns, we feed the local patches at the same spatial position into the proposed Local Sequence Transformer (LST) for further temporal encoding, i. e., the input of our LST is a set like $\mathbf{s}^i = \{\mathbf{x}^{t,i} \in \mathbb{R}^{C \cdot P^2} | t = 1, 2, ..., T\}$, where $i \in \{1, 2, ..., N\}$, and $\mathbf{s}^i$ includes frame patches at the spatial location of $i$.

We show the details of the LST in the middle of Fig. 1, which is divided into two parts: the Local Sequence Embedding and the Low-level Enhanced Transformer stages. In both parts, low-level temporal enhancement is consistently introduced by 3D convolutions given three intuitions: 1) Learnable shallow filters would be better than hand-crafted filters in capturing low-level information in complex situations. 2) Voxels at one specific spatial location may not be well aligned temporally considering camera movements. Thus 3D filters which covers 3D neighborhood structures are more suitable than 2D ones when handling this kind of situation. 3) The patch embedding in Transformers always projects the patch as a whole without more fine-grained locality emphasis. We make up for this using shallow convolutions to enhance low-level feature learning at multiple scales.

**Local Sequence Embedding**. In addition to commonly used linear patch embedding, we involve 3D convolutions at the beginning to enhance low-level temporal modeling. More formally, we define this part as follows:

$$\mathbf{zs}_0^i = [\mathrm{E}_s \mathbf{x}^{1,i}; \mathrm{E}_s \mathbf{x}^{2,i}; ...; \mathrm{E}_s \mathbf{x}^{T,i}], \quad \mathrm{E}_s \in \mathbb{R}^{D \times (C \cdot P^2)} \tag{1}$$

$$[\mathbf{y}_0^{1,i}; \mathbf{y}_0^{2,i}; ...; \mathbf{y}_0^{T,i}] = \mathrm{Maxpool}(\mathrm{Conv3d}([\mathbf{x}^{1,i}; \mathbf{x}^{2,i}; ...; \mathbf{x}^{T,i}]); k), \tag{2}$$

$$\mathbf{zt}_0^i = [\mathrm{E}_0 \mathbf{y}_0^{1,i}; \mathrm{E}_0 \mathbf{y}_0^{2,i}; ...; \mathrm{E}_0 \mathbf{y}_0^{T,i}], \quad \mathrm{E}_0 \in \mathbb{R}^{D \times (C_t \cdot (P/k)^2)} \tag{3}$$

$$\mathbf{z}_0^i = [\mathbf{x}_{temp}^i; \mathbf{zs}_0^i(1) \cdot \sigma(\mathbf{zt}_0^i(1)); \mathbf{zs}_0^i(2) \cdot \sigma(\mathbf{zt}_0^i(2)); ...; \mathbf{zs}_0^i(T) \cdot \sigma(\mathbf{zt}_0^i(T))] + \mathrm{E}_{pos}$$

$$= [\mathbf{z}_{temp}^i; \mathbf{z}^{1,i}; \mathbf{z}^{2,i}; ...; \mathbf{z}^{T,i}], \quad \mathbf{x}_{temp}^i \in \mathbb{R}^D, \quad \mathrm{E}_{pos} \in \mathbb{R}^{(T+1) \times D} \tag{4}$$

where in Eq. (1), $\mathrm{E}_s$ is a trainable linear projection with dimension of $D$, and so is $\mathrm{E}_0$ in Eq. (3), but the difference is that $\mathrm{E}_0$ acts on patches after temporal filtered in Eq. (2), i.e., the enhanced patch

representations at the initial stage $\mathbf{y}_0^{t,i}$. The $k$ in Eq. (2) denotes the pooling kernel size, which is always set to 2 for multi-granularity locality emphasis as discussed previously. The pooling also leads to more efficient training. And $C_t$ in Eq. (3) is the number of the used temporal filters in Conv3d, which is set to 64. Finally, the embedding is given by Eq. (4), where $\mathbf{x}_{temp}^i$ is a learnable temporal token, $\sigma$ indicates sigmoid function, $\mathbf{z}^{t,i} = \mathbf{z}\mathbf{s}_0^i(t) \cdot \sigma(\mathbf{z}\mathbf{t}_0^i(t))$ represents the patch sequence embedding of spatial location $i$ at timestamp $t$. The learnable position embedding $\mathrm{E}_{pos}$ is also kept, but with different meaning of temporal indicator.

**Low-level Enhanced Transformer stage**. Then, patch sequence embeddings at different spatial locations are independently fed into multiple Low-level Enhanced Transformer stages (LET-stages) for further temporal modeling. Similar to the 3D convolutions we used before, we propose to enhance the temporal modeling of patch sequence with aids of shallow spatial-temporal convolution at multiple scales. Given one Transformer block (Trans) [17] defined as:

$$\mathrm{Trans}(\varepsilon) = \mathrm{MLP}(\mathrm{LN}(\varepsilon')) + \varepsilon', \quad \varepsilon' = \mathrm{MSA}(\mathrm{LN}(\varepsilon)) + \varepsilon, \tag{5}$$

where MSA and LN denote multiheaded self-attention and LayerNorm, respectively. With the input embeddings at stage $l-1$: $\mathbf{z}_{l-1}^i = [\mathbf{z}_{temp}^i; \mathbf{z}^{1,i}; \mathbf{z}^{2,i}; ...; \mathbf{z}^{T,i}]$, and the enhanced patch representations: $[\mathbf{y}_{l-1}^{1,i}; \mathbf{y}_{l-1}^{2,i}; ...; \mathbf{y}_{l-1}^{T,i}]$ , we formally define one LET-stage as:

$$\mathbf{z}\mathbf{s}_l^i = [\mathbf{z}_{temp}^{i'}; \mathbf{z}_s^{1,i}; \mathbf{z}_s^{2,i}; ...; \mathbf{z}_s^{T,i}] = \mathrm{Trans}^3(\mathbf{z}_{l-1}^i), \tag{6}$$

$$[\mathbf{y}_l^{1,i}; \mathbf{y}_l^{2,i}; ...; \mathbf{y}_l^{T,i}] = \mathrm{Maxpool}(\mathrm{Conv3d}([\mathbf{y}_{l-1}^{1,i}; \mathbf{y}_{l-1}^{2,i}; ...; \mathbf{y}_{l-1}^{T,i}]); k), \tag{7}$$

$$\mathbf{z}\mathbf{t}_l^i = [\mathrm{E}_l \mathbf{y}_l^{1,i}; \mathrm{E}_l \mathbf{y}_l^{2,i}; ...; \mathrm{E}_l \mathbf{y}_l^{T,i}], \quad \mathrm{E}_l \in \mathbb{R}^{D \times (C_t \cdot (P/k^{(l+1)})^2)} \tag{8}$$

$$\mathbf{z}_l^i = \mathrm{LET\text{-}stage}(\mathbf{z}_{l-1}^i) = [\mathbf{z}_{temp}^{i'}; \mathbf{z}\mathbf{s}_l^i(1) \cdot \sigma(\mathbf{z}\mathbf{t}_l^i(1)); \mathbf{z}\mathbf{s}_l^i(2) \cdot \sigma(\mathbf{z}\mathbf{t}_l^i(2)); ...; \mathbf{z}\mathbf{s}_l^i(T) \cdot \sigma(\mathbf{z}\mathbf{t}_l^i(T))]$$

$$= [\mathbf{z}_{temp}^{i'}; \mathbf{z}^{1,i'}; \mathbf{z}^{2,i'}; ...; \mathbf{z}^{T,i'}], \tag{9}$$

where $\mathrm{Trans}^3$ in Eq. (6) indicates cascaded stacking three blocks together. Overall, the proposed LST is formed by a Local Sequence Emebedding stage and three cascaded Low-level Enhanced Transformer stages.

**Discussion**. Compared with our designs, one *straight thought* might be "just leave the work to self-attention", since theoretically patches can progressively find the most relative patches at the same spatial room for temporal modeling. However, this is nearly impracticable considering both short and long span temporal information is important to our task [65]. Also, the self-attention operation has a quadratic complexity with respect to the number of patches. In contrast, by independently modeling the patch sequences with weight-shared LST, we not only reduce the time complexity of the *straight throught* from $\mathcal{O}(T^2 \cdot N^2)$ to $\mathcal{O}(T \cdot N^2)$, but also explicitly avoid semantic modeling of features like facial structure.

### 3.3 Cross-Patch Inconsistency loss and Cross-Patch Aggregation

After modeling the temporal relation of sequential patches in LST, we have a set of temporal embeddings of all spatial locations $\{\mathbf{z}^i | i = 1, 2, ..., N\}$. However, how to give a final decision is still nontrivial, since the embeddings at different spatial location would represent different levels of temporal changes. For example, pixel variation in the background region is usually less dramatic than that of the mouth region. In addition, there normally exist non-edited regions (usually the background) in deepfakes. Thus, directly adopting an overall binary classification loss on all patches is not the best choice. Instead, we propose to identify the inconsistency by global contrast, because forgery parts should retain heterogeneous temporal patterns compared with the real ones. Concretely, we achieve this goal through the Cross-Patch Inconsistency loss and the proposed Cross-Patch Aggregation.

**Cross-Patch Inconsistency loss**. Given the feature set after LST $\{\mathbf{z}^i | i = 1, 2, ..., N\}$, we leave the first temporal token at all spatial locations ($\{\mathbf{z}_{temp}^i \in \mathbb{R}^D | i = 1, 2, ..., N\}$) for latter classification, and use the remaining patch embeddings ($\{\mathbf{z}^{t,i} \in \mathbb{R}^D | t = 1, 2, ..., T; i = 1, 2, ..., N\}$) in this part (see Fig. 1). We first reduce the temporal dimension of all local regions as: $\mathbf{f}^i = \frac{1}{T} \sum_{t=1}^{T} \mathbf{z}^{t,i}$. Then we calculate the dense cosine similarities of all regions as:

$$\mathbf{sim}^{p,q} = \frac{\langle \mathbf{f}^p, \mathbf{f}^q \rangle}{\|\mathbf{f}^p\| \cdot \|\mathbf{f}^q\|}, \quad p = 1, 2, ..., N; q = 1, 2, ..., N. \tag{10}$$

The sequence of real regions will certainly depict a "natural" variation, while the sequence of fake regions formed by re-assembling will be different. According to the simplest thought that temporal features of real regions should be similar to the real ones, and vice versa. We propose to impose a intra-frame contrastive supervision as:

$$\mathcal{L}_{CPI} = \sum_{p,q} \max\left(\left|\mathbf{sim}^{p,q} - \mathbf{sim}_{gt}^{p,q}\right| - \mu, 0\right)^2, \tag{11}$$

where $\mu$ is a tolerance margin which we set it to 0.1, that allows more diverse feature representations in binary classes. And $\mathbf{sim}_{gt} \in \mathbb{R}^{N \times N}$ is the ground truth similarity matrix that generated from the modification mask sequence $\mathrm{m}_o \in \mathbb{R}^{T \times H \times W}$ as follows:

$$\mathrm{m}_\alpha = \frac{1}{T}\sum_{t=1}^{T} \mathrm{m}_o^t \in \mathbb{R}^{H \times W}, \ \mathrm{m}_\beta = \mathrm{Interpolate}(\mathrm{m}_\alpha) \in \mathbb{R}^{\sqrt{N} \times \sqrt{N}}, \ \mathrm{m} = \mathrm{Flatten}(\mathrm{m}_\beta) \in \mathbb{R}^N,$$
$$\tag{12}$$

$$\mathbf{sim}_{gt}^{p,q} = 2 \cdot (1 - |\mathrm{m}^p - \mathrm{m}^q|) - 1, \quad p = 1, 2, ..., N; q = 1, 2, ..., N. \tag{13}$$

The modification masks $\mathrm{m}_o$ are generated by simply subtracting the fake frame from the corresponding real one, which should be normalize to range of $(0, 1)$ in advance. Thus, the value range of $\mathbf{sim}_{gt}$ is consistent with cosine similarity, i.e., $(-1, 1)$. As for the real clip without mask, the $\mathbf{sim}_{gt} = \mathbf{1}^{N \times N}$.

**Cross-Patch Aggregation**. For the final classification, we want to determine the final prediction based on the overall consistency. We first insert a class token $\mathbf{x}_{class} \in \mathbb{R}^D$ in to temporal tokens of LST output as $[\mathbf{x}_{class}; \mathbf{z}_{temp}^1; \mathbf{z}_{temp}^2; ...; \mathbf{z}_{temp}^N]$. Based on those location-specific temporal tokens, we then adopt additional three Transformer blocks (similar to Eq. (6)) to achieve cross-patch temporal information aggregation. Then the final prediction is given by a fully connection layer taking the [class] embedding as input. With such practice, our model is empowered to identify the different correlations between real and fake sequences. For the classification loss, we simply use the binary cross-entropy loss denoted as $\mathcal{L}_{BCE}$.

And finally, we can train our LTTD in an end-to-end manner with the two losses as:

$$\mathcal{L} = \mathcal{L}_{BCE} + \lambda \cdot \mathcal{L}_{CPI}, \tag{14}$$

where the parameter $\lambda$ is used to balance the two parts and empirically set to $10^{-3}$.

# 4 Experiments

## 4.1 Setups

**Datasets**. Our experiments are conducted based on several popular deepfake datasets including FaceForensics++ (**FF++**) [45], DeepFake Detection Challenge dataset (**DFDC**) [15], CelebDF-V2 (**CelebDF**) [35], FaceShift dataset (**FaceSh**) [30], and DeeperForensics dataset (**DeepFo**) [24]. FF++ (HQ) is used as train set and the remaining four datasets are used for generalization evaluation. FF++ is one of the most widely used dataset in deepfake detection, which contains 1000 real videos collected from Youtube and 4000 fake videos generated by four different forgery methods including Deepfakes [1], FaceSwap [2], Face2Face [53] and NeuralTextures [52]. To simulate the real-world stream media environment, FF++ also provides three versions with different compression rates, which are denoted by raw (no compression), HQ (constant rate quantization parameter equal to 23), and LQ (the quantization parameter is set to 40), respectively. Based on the 1000 real videos of FF++, FaceSh is a later published dataset containing 1000 fake videos, which are generated by a more sophisticated face swapping technique. DeepFo is a large-scale dataset for real-world deepfake detection. To ensure better quality and diversity, the authors make the source videos in a controlled scenario with paid actors. More impressively, a new face swapping pipeline considering temporal consistency is proposed to generate deepfakes with more "natural" low-level temporal features. DFDC is a million-scale dataset used in the most famous deepfake challenge [3]. Following previous works [21], we use more than 3000 videos in the private test set for cross-dataset evaluation in this paper. In addition, CelebDF is one of the most challenging dataset, which is generated using an improved deepfake technique based on videos of celebrities.

**Data preprocessing**. All the used datasets are published in a full-frame format, thus, most of the deepfake detection methods will crop out the face regions in advance. However, with a motivation

to learn the low-level temporal patterns, cropping face regions in advance will lead to artificially introduced jittering due to the independent face detection. Therefore, in our method, we crop the face regions using the same bounding box (including all facial regions) after randomly determine the clip range on-the-fly, where the box is detected by MTCNN [61].

**Implementation details**. The spatial input size $H \times W$ and patch size $P$ is set to $224 \times 224$ and 16, respectively. It is worth noting that the division into such small patches has considerably suppress the overall semantic features. The embedding dimension $D$ is set to 384. In Local Sequence Embedding stage, we use "conv1, conv2_x" of [22] for low-level feature embedding. And we use one $\mathrm{Conv3d}$ layer with kernel size of $3 \times 3 \times 3$ in each LST. For temporal dimension $T$, we empirically set it to 16 and provide more discussion in ablations. We use only the first 128 frames of each video in the experiments, thus the final prediction is averaged from 8 clips. For optimizer, we use Adam [27] with the initial learning rate of $10^{-4}$. When the performance no longer improve significantly, we gradually decay the learning rate. Four NVIDIA A100 GPUs are used in our experiments.

**Evaluation metrics**. Following previous works [45, 21, 31], we use binary classification accuracy (ACC) and Area Under the Receiver Operating Characteristic Curve (AUC) as evaluation metrics.

## 4.2 Generalizability evaluation

For practical application, generalizability should be one of the most concerned properties. Nevertheless, it is usually the Achilles' heel of most deepfake detectors. Since deepfakes generated by different forgery methods (in different datasets) hold different kinds of forgery cues, and overfitting on semantic visual artifacts of the train set can easily lead to cross-dataset evaluation collapse. As we show the performance of models trained on FF++ and tested on four unseen datasets in Table. 1, many methods do not perform satisfactorily. In contrast, our approach outperforms all the recently published novel detectors, and achieves a new state of the art of 91.9 AUC% averaged from the four datasets. Note PatchForensics also focuses on local patches, but only narrows the perceptive field by truncating the CNN without considering the relations between patches globally, it shows limited generalizability.

In addition, FTCN-TT reports comparable results, where they also leverage the self-attention mechanism. But different from our low-level temporal view, they use the semantic features of the whole frame for prediction. Thus, when the visual artifacts are less distinguishable in CelebDF and DFDC, our method surpasses it by a clear margin. A similar situation in LipForensics, which is proposed to perform deep-fake detection using pretrain priors [41] of high-level semantic understanding. With temporal regularity considered, they also achieve comparable results but show suboptimal performance on CelebDF and DFDC. Moreover, PCL+I2G and Face X-ray also focus on low-level learning. However, without considering temporal properties, they exhibit insufficient robustness and perform poorly in DFDC, where the videos are filmed under very different circumstances and have been perturbed to some extent.

Table 1: **Generalizability evaluation**. Models are trained on FF++, and test on remaining four datasets. We show the metric of video-level AUC% comparing with the state of the arts.

| Method | CelebDF | DFDC | FaceSh | DeepFo | *Average* |
|---|---|---|---|---|---|
| CNN-GRU [46] | 69.8 | 68.9 | 80.8 | 74.1 | 73.4 |
| Multi-task [42] | 75.5 | 68.1 | 66.0 | 77.7 | 71.9 |
| PatchForensics [9] | 69.6 | 65.6 | 57.8 | 81.8 | 68.7 |
| FWA [34] | 69.5 | 67.3 | 65.5 | 50.2 | 63.1 |
| Face X-ray [31] | 79.5 | 65.5 | 92.8 | 86.8 | 81.2 |
| PCL+I2G [64] | **90.0** | 67.5 | - | **99.4** | 85.6 |
| SBI+EB4 [48] | 89.9 | 74.9 | 97.4 | 77.7 | 85.0 |
| LipForensics [21] | 82.4 | 73.5 | 97.1 | 97.6 | 87.7 |
| FTCN-TT [65] | 86.9 | 74.0 | 98.8 | 98.8 | 89.6 |
| LTTD (ours) | 89.3 | **80.4** | **99.5** | 98.5 | **91.9** |

## 4.3 Robustness to perturbations

Another vital property for practical application is robustness. For network transmission, videos are always compressed. Also, consider possible attacking against the detectors, we evaluate our approach on different types of perturbed videos.

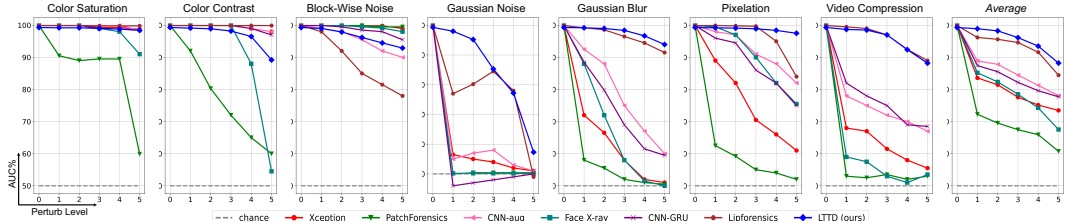

Figure 2: **Robustness evaluation**. Models are trained on the the *clean* train set of FF++ and tested on perturbed test sets, respectively. *Average* indicates the results averaged from all types of perturbations.

Following previous works [21, 24], we use the script [2] with FF++ and generate seven types of perturbations at five levels. As we show the diagram in Fig. 2 (others refer to [21]), the seven perturb methods include color saturation change, color contrast change, block-wise noise, gaussian noise, gaussian blur, pixelation, and video compression. Examples can be found here.

Comparing with other methods, our approach shows excellent robustness against most of the perturbations. As for the type of gaussian noise at a relatively higher level, the underlying low-level patterns are severely disturbed that our method can not work properly. We also present the comparison with two state of the arts in Table. 2, where the results are averaged from all five levels. In general, Face X-ray detects low-level boundary artifacts, and LipForensics focuses on high-level semantic features. Thus, the former performs better with block-wise noise type and the latter shows better results with others. In contrast to Face X-ray, we detect low-level patterns in spatial-temporal space, which is better resistant to perturbations, and thus achieves better performance.

Table 2: **Robustness evaluation**. Average performance evaluated on perturbed videos at five levels. Clean: origin videos, CS: color saturation, CC: color contrast, BW: block-wise noise, GNC: gaussian noise, GB: gaussian blur, PX: pixelation, VC: video compression, Avg: averaged performance on distored videos, Drop: performance drop comparing to Clean. The gray numbers do not reflect robustness, and metrics of video-level AUC% is reported.

| Method | Clean | CS | CC | BW | GNC | GB | PX | VC | *Avg/Drop* |
|---|---|---|---|---|---|---|---|---|---|
| Face X-ray [31] | 99.8 | 97.6 | 88.5 | **99.1** | 49.8 | 63.8 | 88.6 | 55.2 | 77.5/-22.3 |
| LipForensics [21] | 99.9 | **99.9** | **99.6** | 87.4 | 73.8 | 96.1 | 95.6 | **95.6** | 92.6/-7.3 |
| LTTD (ours) | 99.4 | 98.9 | 96.4 | 96.1 | **82.6** | **97.5** | **98.6** | 95.0 | **95.0/-4.3** |

## 4.4 Ablations

**Module effects**. We first compare our method with related baselines and several alternative designs. 1) **Xception** is the commonly used backbone in deepfake detection; 2) **ViT** is the most famous vision transformer backbone, where we use the "small" version with embedding dimension of 384; 3) **ViViT** [8] is a recently published work developed in the self-attention style with spatio-temporal modeling ability for action recognition. We use the best model, "Model 1", evaluated in their paper with a comparable backbone, "ViT small", with our method; 4) **LTTD w/o LST** indicates the model that we replace the proposed LST with commonly used Patch Embedding and Transformer blocks [17]; 5) **LTTD w/o CPI** is our LTTD framework trained without using $\mathcal{L}_{CPI}$ (Eq. (13)); 6) **LTTD w/o CPA** represents the model that we replace the CPA module with a simple fully connected classification layer after average pooling the temporal embeddings from all spacial locations. From Table. 3, all the models achieve nearly perfect results on FF++. While in the cross-dataset setting, our model with elaborately designed modules exhibits much stronger generalizabilty, and the three components consistently boost the best results.

**Visualization**. To intuitively demonstrate the reason that our approach is more generalizable, we compare the feature representations before the final classification in Fig. 3. Models are trained on FF++ and tested on four subsets of FF++ (Deepfakes, Face2Face, FaceSwap, and NeuralTextures) respectively. Due to the abundant inductive bias of the convolution, Xception clearly split the four

---

[2] https://github.com/EndlessSora/DeeperForensics-1.0/tree/master/perturbation

Table 3: **Module effect**. Models are trained on FF++. Gray numbers reflect in-dataset effectiveness, and others represent cross-dataset generalization. *Cross-Avg*: average from unseen datasets.

| Method | FF++ | | FaceSh | | DFDC | | DeepFo | | *Cross-Avg* | |
|---|---|---|---|---|---|---|---|---|---|---|
| | ACC% | AUC% | ACC% | AUC% | ACC% | AUC% | ACC% | AUC% | ACC% | AUC% |
| Xception [11] | 96.08 | 99.38 | 72.47 | 78.60 | 60.47 | 67.36 | 69.21 | 83.28 | 67.38 | 76.41 |
| ViT [17] | 95.00 | 97.92 | 62.86 | 65.56 | 64.81 | 72.89 | 71.85 | 83.24 | 66.51 | 73.90 |
| ViViT [8] | 94.71 | 97.92 | 63.21 | 77.40 | 67.52 | 74.16 | 60.12 | 82.86 | 63.62 | 78.14 |
| LTTD w/o LST | 95.57 | 98.57 | 90.71 | 97.44 | 60.40 | 70.06 | 87.68 | 97.94 | 79.60 | 88.48 |
| LTTD w/o CPI | 97.29 | 99.23 | 95.00 | 98.86 | 67.95 | 77.03 | 90.62 | 97.68 | 84.52 | 91.19 |
| LTTD w/o CPA | 96.14 | 99.00 | 91.79 | 99.35 | 65.29 | 70.64 | 87.39 | 96.62 | 81.49 | 88.87 |
| LTTD | 97.72 | 99.52 | 96.55 | 99.51 | 71.34 | 80.39 | 92.53 | 98.50 | 86.81 | 92.80 |

kinds of deepfakes into four clusters, even **only binary labels are used** in training. This shows that the strong CNN tends to overfit on method-specific artifacts generated by different deepfake methods, although plausible results are achieved in in-dataset testing, it is harmful to generalize to unseen deepfakes. In addition, similar phenomena can be found in the results of ViT. However, the divison of the four clusters is not as clear as in Xception due to the lesser inductive biases introduced in ViT. In contrast, the features of our approach show a completely different outline, where the real videos are also clearly separated, but the remaining four types of deepfakes are compacted in a unified manifold. We attribute this phenomenon to our low-level temporal learning, which can distinguish deepfakes from real depending on more fundamental low-level temporal inconsistencies, thus achieving the best generalizability.

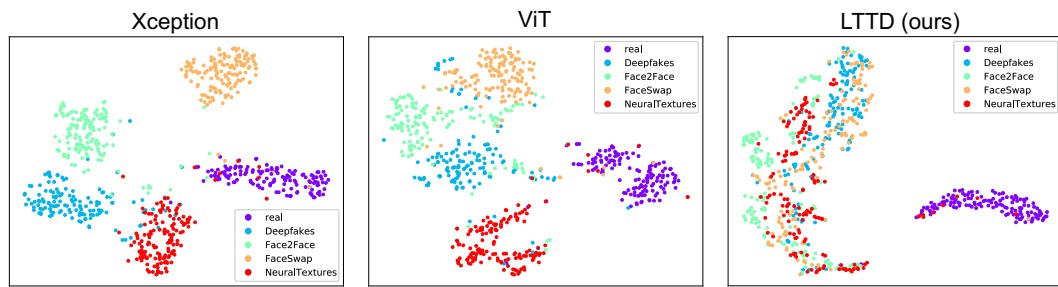

Figure 3: **Feature visualization**. In the t-SNE [54] visualization, every dot represents a compacted features of the corresponding test video, and different color indicates different class.

**Sequential input**. As we mentioned previously, we empirically set the clip length to 16. On the one hand, too short a clip does not ensure enough temporal information, e. g., 4 consecutive frames are almost identical to each other. On the other hand, too long a clip is not necessarily better: 1) A longer clip costs more memory; accordingly, a smaller batch size also slow down training convergence or worse. 2) We found some of the videos in the test datasets contained scene switching. Considering the FPS of 24, a video clip in our experiments will not last 1 second, thus greatly avoiding the scene-switching problem.

Here, we further analyze how clip sampling will affect the low-level temporal learning of the proposed LTTD. Two hyper-parameters are investigated: clip length and frame sampling space. From Table 4, the first 4 rows show that clip length has little effect on the results. We believe this is closely related to the idea of low-level temporal learning, which does not require the video clip to last long enough (e. g., 3 seconds) but only adequate frames to extract the low-level temporal patterns. Sparse sampling is another practice to aggregate more content in a single clip. When we expand the frame interval, the performance degrades considerably. This phenomenon suggests that sparse sampling is detrimental to the learning of low-level temporal patterns even when more motion content is included. These

Table 4: **Sequential input ablation**. CL: clip length, FSS: frame sampling space.

| CL | FSS | FF++ | DFDC | DeepFo | *Average* |
|---|---|---|---|---|---|
| 8 | 1 | 99.26 | 77.25 | 96.82 | 91.11 |
| 16 | 1 | 99.52 | 80.39 | 98.50 | 92.80 |
| 32 | 1 | 99.47 | 80.92 | 98.41 | 92.93 |
| 64 | 1 | 99.38 | 79.23 | 98.49 | 92.37 |
| 16 | 2 | 99.15 | 76.17 | 97.32 | 90.88 |
| 16 | 4 | 98.51 | 73.02 | 91.33 | 87.62 |

findings demonstrate that LTTD is distinctly different from related temporal-based models, since the low-level temporal learning is specially designed for deepfake detection.

**Forgery localization**. Our model enjoys a local-to-global learning protocol, where differences between real and fake regions are naturally explored. Here we investigate this property by visualizing the CAM [47] responses of our model. Considering the input size of $224 \times 224$ and the patch size of $16 \times 16$, the spatial space is divided into a $14 \times 14$ grid. We use the CAM responses of the $x_{class}$ token to draw a localization map by bilinear interpolation.

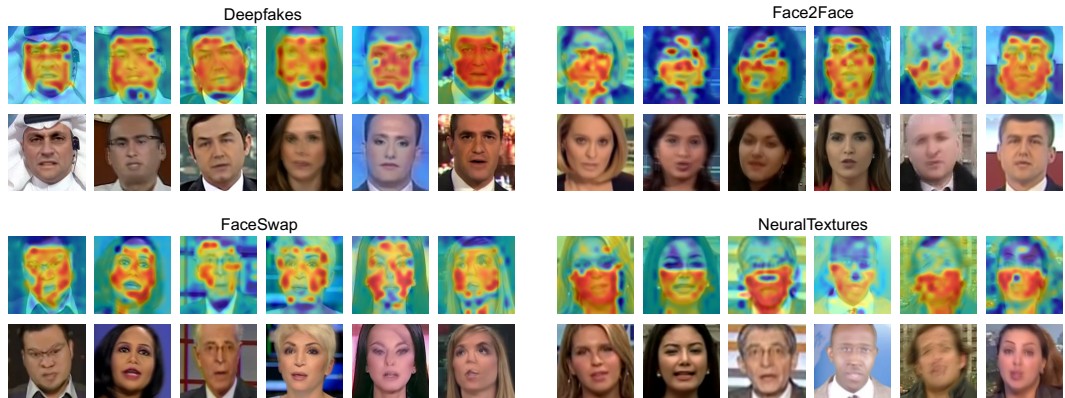

Figure 4: **Forgery localization**.

As shown in the Fig. 4, our model is able to identify the local inconsistencies. In addition, we can learn some different characteristics of each forgery type. Most distinctively, regions of mouth and eyes in FaceSwap are not modified, thus showing patterns that distinguish from other facial parts. Moreover, regions relating to the mouth are reenacted in NeuralTextures, just like the localization results shown in the figure.

Despite the good intuitive demonstrations, it remains future works to determine whether the localization results are credible.

# 5 Conclusion and discussion

**Conclusion**. In this paper, we propose a reliable framework to address the practical problems of deepfake detection, which emphasizes on the low-level temporal patterns of sequential patches in the restricted spatial region with a whole-range temporal receptive field using Transformer blocks. In addition, we make the final classification in a more general global-contrastive way. Thus better generalizability and robustness are achieved to better support deepfake detection in real-world scenes. Moreover, qualitative results further verify that low-level temporal information can lead to stronger generalizability, which could also be a guideline for developing better approaches in the future.

**Limitation**. Considering the continuous advances in deepfake creation and adversarial training, the performance of our approach when encountering low-level and temporal adversarially enhanced deepfakes in the future is yet unclear. Moreover, although our method shows favorable performance compared to recent works, it still requires intensive labor in order to handle real-world scenarios. This is a commonly shared limitation that we do not know if the detectors are *calibrated well* for real-world deployment. In addition to identifying deepfakes, how we can ensure the predictions are credible remains an open problem, hindering the application of all deepfake detectors.

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
