# Delving into Sequential Patches for Deepfake Detection

# 1 Appendix

## 1.1 Dataset details

FF++ is one of the most widely used dataset in deepfake detection, which contains 1000 real videos collected from Youtube and 4000 fake videos generated by four different forgery methods including Deepfakes [1], FaceSwap [2], Face2Face [8] and NeuralTextures [7]. To simulate the real-world stream media environment, FF++ also provides three versions with different compression rates, which are denoted by raw (no compression), HQ (constant rate quantization parameter equal to 23), and LQ (the quantization parameter is set to 40), respectively. Based on the 1000 real videos of FF++, FaceSh is a later published dataset containing 1000 fake videos, which are generated by a more sophisticated face swapping technique. DeepFo is a large-scale dataset for real-world deepfake detection. To ensure better quality and diversity, the authors make the source videos in a controlled scenario with paid actors. More impressively, a new face swapping pipeline considering temporal consistency is proposed to generate deepfakes with more "natural" low-level temporal features. DFDC is a million-scale dataset used in the most famous deepfake challenge [3]. Following previous works [4], we use more than 3000 videos in the private test set for cross-dataset evaluation in this paper. In addition, CelebDF is one of the most challenging dataset, which is generated using an improved deepfake technique based on videos of celebrities.

## 1.2 Data preprocessing details

As the goal of our method is to learn the low-level temporal features, the *per-frame face cropping* applied by most works is not optimal for our consideration. Instead, whether for *fake* or *real* video, keeping the original voxel-level changes should be a precondition for our model input. We thus crop the faces using a same bounding box to form a clip where the same box should lastingly cover the face. This processing can also be learned from the following steps:

- Given a full-frame video.

- Randomly determine a valid clip range, e.g., from frame 10 to frame 25.

- Sample the determined clip with 16 successive frames.

- Detect all the bounding boxes of the faces in the 16 frames, resulting in 16 boxes (case of multi-face is omitted here).

- Generated the ***crop-box*** by picking the largest or smallest indices on each coordinate as:

```python
# boxes.shape = [16, 4]
# [xmin, ymin, xmax, ymax] = boxes[0]
box = boxes.min(axis=0)[:2].tolist() +
      boxes.max(axis=0)[2:].tolist()
```

- (Optional) Expand or narrow the ***crop-box***. In this paper, the ***crop-box*** is enlarged by a factor 0.15.

- Crop the faces using the ***crop-box*** to form a face clip as model input.

## 1.3 Low-level enhancement implementation

In terms of the low-level enhancement, we include a detailed torch-like description here to demonstrate how shallow 3D filter works:

```python
# x: LST features, [B*patch_num, T+1, embed_dim]
# y: enhanced features of last stage
# p: patch size of last stage
# conv3d: 3D filer, kernel_size=3x3x3
# patch_embed: linear transform

y = F.max_pool3d(y, kernel_size=(1, 2, 2), stride=(1, 2, 2))
B, C, T, H, W = y.shape
p = p // 2

y = F.unfold(y.transpose(1, 2).reshape(B * T, C, H, W), kernel_size=(p
                            , p), stride=(p, p)).view(B, T, C,
                            p, p, -1).permute(0, 5, 2, 1, 3, 4)
                            .view(-1, C, T, p, p)
st_feat = conv3d(y)
st_feat = st_feat.view(B, H // p, W // p, -1, T, p, p).permute(0, 3, 4
                            , 1, 5, 2, 6).contiguous().view(B,
                            -1, T, H, W).transpose(1, 2).
                            contiguous().view(B * T, -1, H, W)
st_feat = patch_embed(st_feat)
BT, patch_num, embed_dim = st_feat.shape
st_feat = st_feat.view(B, T, patch_num, embed_dim).permute(0, 2, 1, 3)
                            .reshape(-1, T, embed_dim)

temp_token = x[..., 0, :]
x_enhance = torch.zeros_like(x)
x_enhance[..., 1:, :] = x[..., 1:, :] * st_feat.sigmoid()
x_enhance[..., 0, :] = temp_token
x = x_enhance
```

## 1.4 Robustness ablation

Table 1: **Robustness ablations**. Average performance evaluated on perturbed videos at five levels. Clean: origin videos, CS: color saturation, CC: color contrast, BW: block-wise noise, GNC: gaussian noise, GB: gaussian blur, PX: pixelation, VC: video compression, Avg: averaged performance on distored videos, Drop: performance drop comparing to Clean. The gray numbers do not reflect robustness, and metrics of video-level AUC% is reported.

| Method | Clean | CS | CC | BW | GNC | GB | PX | VC | *Avg/Drop* |
|---|---|---|---|---|---|---|---|---|---|
| Face X-ray [5] | 99.8 | 97.6 | 88.5 | **99.1** | 49.8 | 63.8 | 88.6 | 55.2 | 77.5/-22.3 |
| LTTD w/o LST | 98.8 | 93.9 | 92.6 | 86.6 | 68.8 | 93.2 | 95.9 | 91.7 | 88.9/-9.9 |
| LTTD w/o CPI | 98.8 | 93.9 | 92.6 | 86.2 | 68.9 | 93.2 | 95.1 | 91.7 | 88.8/-10.0 |
| LTTD w/o GCC | 99.1 | 97.6 | 90.6 | 94.9 | 76.0 | 89.0 | 97.4 | 91.4 | 90.9/-8.2 |
| LTTD | 99.4 | 98.9 | 96.4 | 96.1 | 82.6 | 97.5 | 98.6 | 95.0 | 95.0/-4.3 |

We conduct ablations for generalization and here we provide more discussion with the robustness. From the results in Table 1, we find that the special designs all contribute to optimal performance. For color contrast (CC), the Global Contrastive Classification (GCC) module has a more significant contribution as it better enhances the detection of local color anomalies by modeling features in different spatial regions through global comparisons. In contrast to block-wise noise (BW), the Local Sequence Transformer (LST) and Cross-Patch Inconsistency (CPI) modules contribute more, since BW noise affects only a very small local area, it has no effect on the low-level temporal features in other regions, however, the global contrast learning of GCC will be interfered. The results on gaussian

noise (GNC) can be understood consistently with BW. Since GNC comprehensively modifies the low-level features, low-level temporal learning of LST and CPI will be greatly affected, while the global contrastive learning of GCC is less affected, thus leading to a more significant contribution of GCC. Compared with Face X-ray focusing on spatial low-level feature learning, the performance degradation of our models are significantly smaller due to the consideration of temporal dimension. This phenomenon is also in line with our motivation we discussed in the Sec. Introduction that low-level features are susceptible to perturbations and robustness will be enhanced by incorporating temporal learning.

### 1.5 Forgery localization

Our model enjoys a local-to-global learning protocol, where differences between real and fake regions are naturally explored. Here we investigate this property by visualizing the CAM [6] responses of our model. Considering the input size of $224 \times 224$ and the patch size of $16 \times 16$, the spatial space is divided into a $14 \times 14$ grid. We use the CAM responses of the $x_{class}$ token to draw a localization map by bilinear interpolation.

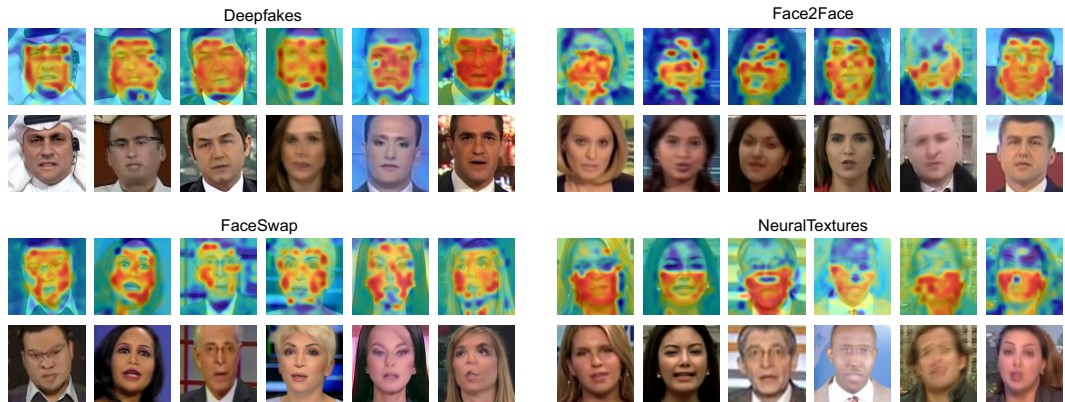

Figure 1: **Forgery localization**.

As shown in the Fig. 1, our model is able to identify the local inconsistencies. In addition, we can learn some different characteristics of each forgery type. Most distinctively, regions of mouth and eyes in FaceSwap are not modified, thus showing patterns that distinguish from other facial parts. Moreover, regions relating to the mouth are reenacted in NeuralTextures, just like the localization results shown in the figure.

Despite the good intuitive demonstrations, it remains future works to determine whether the localization results are confidently credible.