# OpenReview forum: "Delving into Sequential Patches for Deepfake Detection"
_NeurIPS.cc/2022/Conference — NeurIPS 2022 Accept_

### Official Review · Reviewer_DdUQ · 2022-07-08

**Rating:** 5
**Confidence:** 4
**Soundness:** 3 good
**Presentation:** 2 fair
**Contribution:** 2 fair

**Summary:**

The paper proposes a method for DeepFake Detection. The method focuses on a particular way of modeling the temporal aspect of the detection: instead of learning first spatio-temporal downsampled embedding with 3D convolution and then aggregating the embedding with transformers [62], the proposed method immediately breaks the image into patches and adds the temporal dimension. Unlike prior work, temporal modeling is done directly over patches with the motivation of identifying low-level temporal inconsistencies and standing on the shoulders of insights from [35]. This is achieved by moving the transformer architecture immediately upstream in the pipeline contrary to previous methods in which usually temporal aggregation is performed downstream. Though the method models patches in this way still use part of 3D convolutions applied to patches. The method is supervised with standard cross-entropy loss and another loss that enforces that real patches should be similar to each other in the learned embedding space.
Experimental results are provided on recent benchmarks for Deepfake detection along with ablation studies. The robustness of the method to common perturbation is also tested.

**Questions:**

- Tab. 3 maybe improve if an average is presented.
- No detail of training and testing time.
- What are the parameters of the 3D Convolution used? Kernel size over spatial dimension and temporal.

- There are several parts that are not clear:
   > L198: the simplest thought is that the real region should be similar to the real one to some extent

For me, this is not enough to justify the loss and is not a very clear sentence. The sentence says that real regions should be similar to real ones but this is not true in the image space; maybe this is the requirement that the loss is imposing in the learned feature space though is not clear. Furthermore, it is not clear what the loss why the loss matches the cosine similarity between all patches with the ground-truth similarity matrix. Which min and max values take the ground-truth similarity matrix?
The process of generating the ground-truth similarly matrix is displayed in Eq. (12,13) but seems over complicated with the notation ($m_\alpha,m_o,m_\beta,$). Why does interpolation have to be $\pi$? Per my understanding, the min and max values for sim_gt are $\{-1, 1\}$. Is this correct? Anyway, the motivation for the loss is that real features should be basically similar to each other but it is not clear how this is attained given the current loss formulation since there is no selection of the real embedding in the loss unless this is achieved with the $sim_git$ though how?

 L:208 $\mathbf{x}_{class}$ is mentioned without explaining what is it.

L:242, In the same way, I mentioned in the weakness the part on data processing is hard to understand. Especially this sentence:
>  Therefore, in our method, we crop the face regions using the same bounding box after randomly determining the clip range in-the-fly,





**Limitations:**

The limitations presented are somewhat related to the generalization of the model. It says _now it is good (at least a bit superior to the state-of-the-art) but we will be unsure in the future._

I think an overall direction for the future that would be nice for Deepfake Detection is working toward ensuring that the prediction of the Detection is really calibrated well. To the best of my knowledge, there is no detection system yet deployed that can be left free of running on video on social networks and flagging videos. I am not sure if there are implementations with humans in the loop that check highly confident video marked by the detector. Anyway, the point is that, given that an erroneous highly confident prediction as fake for an authentic video could be even more problematic of a deepfake itself, all these deepfake detectors have the limitation that we do not know if they are _calibrated well_.  Perhaps adding a general discussion on this may improve the paper: i.e. as of now, the method has good generalization accuracy, but if we run it in the wild on videos for which we do not have labels, how much can we trust the prediction?

**Strengths And Weaknesses:**

### Strenghts
- The method is overall motivated when proposing something and takes a direction that is different than the state-of-the-art [62] of directly modeling patch sequences bringing transformers upstream in the pipeline.
- Overall the innovative claims in the intro are well balanced with the paper's experimental evidence, especially in the robustness part of perturbations.
- I was able to grasp the overall idea of the method quickly though often parts feel overcomplicated in the way they are written.
- The experimental validation is promising and uses recent benchmarks for deepfake detection. Ablation studies are provided to support design choices. In some cases, the ablation improvement is clear in other cases less clear. In Tab. 3 an overall summary using average with standard deviation across unseen manipulation is needed.


### Weaknesses

- The average improvement over the competing method [62] `(89.6% vs 91.8%) `is appreciated but looks a bit marginal without comparing the timings of the approach. Also, it is unclear if the improvement `+2.2` video-level AUC is statistically significant. What happens if you train multiple **LTTD** for different initialization and multiple **FTCN-TT** and report the average and standard deviation? Is the improvement statistically significant, given the standard deviation of the two approaches? (Assuming authors have code to run FTCN-TT). I ask this because, it is clear that from the distribution of performance the gap is obtained thanks to the DFDC dataset, all the others are basically the same. What makes LTTD works well on DFDC? This small improvement questions the overcomplexity of the method and I have the idea that the paper will be hard to reproduce and re-implement.
-  Though the method is well-motivated, it feels a bit the method is riding the success of transformers and applied to the idea of modeling temporal discrepancy with patches for DeepFake Detection, changing the idea of [62]. Also, though the method seeks a changing paradigm does still apply 3D convolution as in [62]. This part where the method applies 3D convolution on patches is less clear. Convolution works on images because of the assumption that patches can be highly correlated but here the method breaks the image by splitting into patches, so more explanation on this 3D convolution is needed for the way it handles the spatial component over patches.
- The idea to break the image in patches does not account for rigid and non-rigid deformation that the face may even in 3D, convolution suffers from this too but is more resistant to jittering of translation by definition. The paper says something on this matter at `L149` but is not clear.
- `L242` gives details about the face cropping which is related to alignment, motion compensation, and breaking the image into patches. It is not clear overall how the face cropping of the system works. The method does not do frame-by-frame face detection and cropping but uses a single crop. The text says using `the same bounding box` but same respect to which frame? It is not clear also what is randomly determined. If a single crop is used and the face move slightly this is even worse than the random jittering that you get by frame-by-frame processing.
- More on this: In section 3.2 it is not clear if the Conv3D still slides over the patches or if the spatial kernel size of the convolution is big as the patch itself. This part spends a lot of space with equations (1-4) with tons of super and subscripts but it does not make the reading easier.
- It is ok to be formal with the notation yet very often the article abuses notation and this makes the article not easily readable.
There are multiple of these remarks e.g. usage of subscript that is not necessary $ \mathbf{x}_{p}^{i}$. Why $p$ is used in L.131, L144 I believe the patch is indexed by $i$. Same remarks for Eq. (5-8).
- Given that the title includes _Delving into Sequential Patches_, the method misses an instructive study of what happens when the temporal dimension $T$ varies from different values than 16; also, given a fixed $T$, the paper misses a study to understand whatever it is better to have a dense sampling of the frames (e.g. 1:1 wrt to the original video) and a small window or maybe more sparse sampling of the frames (1:20) but to aggregate more content of the video.
- Fig. 1 is overall nice and I appreciated but it is pretty dense and not easy to digest immediately.
- I found some typos and sometimes the text use adjective a bit off: e.g. `L17 ` `enormous fake videos`? This means that a video is huge, big in size. `L81` data aRgumentation. `L199` vise versa (vice versa). `L308` a unified manifolds.

### Justification of the rating
My opinionf the paper is that it is a good system paper for deepfake detection with marginal improvement over the state-of-the-art, assuming these are statistically significant. The paper has some strength in showing more robustness than other methods. The overall claims are decently supported. Besides that, the paper needs to be improved by simplifying the notation by making it more clear for the community since as of now it will be hard to reproduce it. Overall it is a good paper but I believe that if the paper is not accepted, the deepfake community is not going to lose innovative ideas. According to this, I sugged a borderline reject score before rebuttal.

---

> ### Author Response · Authors · 2022-08-02
> **Author Response to Reviewer DdUQ (part3/3)**
>
> > **Q10: "L:208 xclass is mentioned without explaining what is it."**
>
> **A10**: $x_{class}\in \mathbb{R}^{D}$ is a learnable token inserted at the beginning of embeddings of sequential patches. It has the same role and meaning as the *class token* in ViT. The embedding of this token is used for final classification.
>
> > **Q11: "The limitations presented are somewhat related to the generalization of the model. It says *now it is good (at least a bit superior to the state-of-the-art) but we will be unsure in the future.* I think an overall direction for the future that would be nice for Deepfake Detection is working toward ensuring that the prediction of the Detection is really calibrated well... all these deepfake detectors have the limitation that we do not know if they are *calibrated well*... if we run it in the wild on videos for which we do not have labels, how much can we trust the prediction? "**
>
> **A11**: We really appreciate and agree with the limitation pointed out by the reviewer about *'calibration requirements*. We believe it will be an important direction of the next generation’s deepfake detection. At the current stage, we still need intensive labor to ensure the detector is *calibrated well* in the deployment. We have included related discussions in the revision and are very grateful for the insightful discussion.
>
> > **Q12: "Justification of the rating: My opinionf the paper is that it is a good system paper for deepfake detection..."**
>
> **A12**: We are grateful that our efforts and ideas are recognized by the reviewer. All the constructive comments from the reviewers lead to a better revised paper! We believe that our encouraging finding of low-level & temporal learning in this paper will be a non-trivial contribution to the deepfake detection community. The novel idea of framework designs and the new state of the art can also promote its development. We hope you can change your mind and let our paper contribute to the community!

---

> ### Author Response · Authors · 2022-08-02
> **Author Response to Reviewer DdUQ (part2/3)**
>
> > **Q4: "More on this: In section 3.2 it is not clear if the Conv3D still slides over the patches or if the spatial kernel size of the convolution is big as the patch itself..."**
>
> **A4**: The 3D convs operators in our work do not slide over the patches. They only focus on the specific spatial region. The kernel size is set to 3x3x3. The patches at different spatial regions would not interact with each other during Local Sequence Transformer (LST) forwarding, and only patches at the same spatial location are encoded by all the operators. We have included a detailed torch-like description of the 3D convs in the revised supplementary material. We have simplified the symbols of the equations for a better demonstration. Please see the revised paper for details.
>
> > **Q5: "It is ok to be formal with the notation yet very often the article abuses notation and this makes the article not easily readable. There are multiple of these remarks e.g. usage of subscript that is not necessary xpi. Why p is used in L.131, L144 I believe the patch is indexed by i. Same remarks for Eq. (5-8)."**
>
> **A5**: Thank you for the valuable suggestions, we find the simplified equations easier to understand. We have revised the related parts and marked them in blue in the revision. The subscript $p$ is used to denote the origin image patch. We also find it dispensable and remove it in the revision.
>
> > **Q6: "... the method misses an instructive study of what happens when the temporal dimension T varies from different values than 16; also, given a fixed T, the paper misses a study to understand whatever it is better to have a dense sampling of the frames (e.g. 1:1 wrt to the original video) ..."**
>
> **A6**: Thank you for the valuable suggestion. We include related discussions in the revision. Please find the details in the "General Responses" and the revised paper.
>
> > **Q7: "I found some typos and sometimes the text use adjective a bit off: e.g. `L17 ``enormous fake videos`? ...  `L81` data agumentation. `L199` vise versa (vice versa). `L308` a unified manifolds."**
>
> **A7**: Thank you for pointing out these typos and the suggestion! We have revised them accordingly.
>
> > **Q8: "No detail of training and testing time."**
>
> **A8**: Our models are trained on one A100 GPU for 12 hours. The inference speed for processed videos is about 98 FPS.
>
> > **Q9: "not clear:  L198: the simplest thought is that the real region should be similar to the real one to some extent".... Which min and max values...? ...Eq. (12,13) but seems over complicated with the notation (mα,mo,mβ,). ... the motivation for the loss is that real features should be basically similar to each other but it is not clear how this is attained given the current loss formulation ...?"**
>
> **A9**:
>
> **1)** Our intention is to demonstrate that low-level temporal features of the real region should be similar to the real ones since the sequence of real regions will certainly depict a "natural" variation, while the sequence of fake regions formed by re-assembling will be different. We include a related explanation in the revised paper.
>
> **2)** About the min and max values of $\mathrm{sim}_{gt}$ are -1and 1, respectively.
>
> **3)** We have revised these equations and fixed some bugs. Here we briefly explain "how ` real features should be basically similar to each other is attained given the current loss formulation":
>    a) Given the modification mask as a gray image generated by simply subtracting the fake frame from the corresponding real one. The part that is faked out will have a larger difference and thus be closer to 255. In contrast, the real part will be closer to 0. Then, we normalize the mask to the value range of (0, 1).
>    b) The Eq (12)  first average pools the mask sequence at the temporal dimension, then interpolate the spatial size to be consistent with the feature map. Finally, we use $\mathrm{m} \in \mathbb{R}^{N}$ to denote the flattened map. Thus, each value of $\mathrm{m}$ corresponds to one patch sequence of the input clip. If the value of $\mathrm{m}$ is closer to 1, it means that there is more forged content at the location of this patch, and vice versa.
>    c) Therefore, by Eq (13), two patches with similar values in the flatten map $\mathrm{m}$ will get a greater ground truth similarity.

---

> ### Author Response · Authors · 2022-08-02
> **Author Response to Reviewer DdUQ (part1/3)**
>
> We sincerely thank the reviewer for the constructive comments. We will respond to each detailed concern as follows:
>
> > **Q1: "The average improvement over the competing method [62] `(89.6% vs 91.8%) `is appreciated but looks a bit marginal without comparing the timings of the approach. Also, it is unclear if the improvement `+2.2` video-level AUC is statistically significant ... This small improvement questions the overcomplexity of the method ..."**
>
> **A1**:
>
> **1)** Compared with FTCN-TT, we have tried to re-implement it ourselves (since no training code is open sourced) but found it is hard to reproduce performances comparable to the original paper. Thus, we directly cited the numbers.
>
> **2)** In terms of the performance of our method. We retrain the model 5 times with different random seeds and get the following results:
>
> | Method | CelebDF        | DFDC           | FaceSh         | DeepFo         |
> | ------ | -------------- | -------------- | -------------- | -------------- |
> | FTCN   | 86.9           | 74.0           | 98.8           | 98.8           |
> | LTTD   | 89.25$\pm$0.13 | 80.39$\pm$0.46 | 99.51$\pm$0.07 | 98.50$\pm$0.21 |
>
> **3)** For model complexity, please refer to the "General Responses".
>
> > **Q2: "Though the method is well-motivated, it feels a bit... changing the idea of [62]... This part where the method applies 3D convolution on patches is less clear ... more explanation on this 3D convolution is needed for the way it handles the spatial component over patches."**
>
> **A2**:
>
> We would like to first explain the 3D convolutions in our paper and then discuss the related method FTCN [57].
>
> **1)** In FTCN, deep layers of 3D convs operate on the whole frames and thus inevitably focuses on semantic motions.
>
> While in our method, the introduced Low-level Enhancement with 3D convolution is motivated by the previous low-level feature learning methods which use hand-crafted filters [40,19,26,35]. Our intuition here is to leverage shallow learnable 3D filters for low-level information processing on local patches. Specifically, one 3d conv layer operates on each stage of the Low-level Enhancement Transformer like a spatio-temporal filter. Moreover, it would never model the cross-patch relations. A detailed torch-like description is included in the revised supplementary material.
>
> **2)** The related work, FTCN, is mainly based on *deep 3D CNNs* to learn pixel-level temporal discrepancy. The self-attention layers in their model are only responsible for multi-frame semantic information aggregation.
>
> We adopt Transformers to model the sequential patches motivated by two facts: a) various practices in computer vision have demonstrated that self-attention can be used directly to model vision content by regarding visual patches as tokens; b) Transformers are not restricted by receptive fields. We thus leverage this property for both long- & short-span temporal learning. While in FTCN and other 3D conv-based works, the temporal receptive field is restricted by the kernel size of 3D convolutions.
>
> Overall, FTCN and our work are related but fundamentally different.
>
> > **Q3:  "... It is not clear overall how the face cropping of the system works... It is not clear also what is randomly determined. If a single crop is used and the face move slightly this is even worse than the random jittering that you get by frame-by-frame processing."**
>
> **A3**: The processing of face crop in our framework is demonstrated as:
>
> 1. Given a full-frame video;
>
> 2. We randomly determine a valid clip range, e.g., from frame 10 to frame 25;
>
> 3. We sample the determined clip with 16 successive frames;
>
> 4. We detect all the bounding boxes of the faces in the 16 frames, resulting in 16 boxes (case of multi-face is omitted here);
>
> 5. `the same bounding box` is generated by picking the largest or smallest indices on each coordinate as:
>
>    ```python
>    # boxes.shape = [16, 4]
>    # [xmin, ymin, xmax, ymax] = boxes[0]
>    box = boxes.min(axis=0)[:2].tolist() + boxes.max(axis=0)[2:].tolist()
>    ```
>
> We also include the details in the revised supplementary material.
>
> Faces in the videos are always slowly moving. Considering the FPS of 24, a video clip in our experiments will not last for 1 second. Such practice greatly avoids large movements in a single clip. Frame-by-frame processing will corrupt the temporal relations of low-level features.

---

### Official Review · Reviewer_Yygr · 2022-07-11

**Rating:** 5
**Confidence:** 4
**Soundness:** 3 good
**Presentation:** 3 good
**Contribution:** 3 good

**Summary:**

The proposed method combines 3D conv and vision transformer to extract forgery artifacts in Deepfakes videos, the experiments prove that the proposed method achieves outstanding generalization performance on multiple Deepfakes datasets.



**Questions:**

1. In the discussion part for Section 3.2, the authors claim that the proposed structure which combines 3D conv feature and self-attention feature can explicitly avoid semantic modeling of features like facial structure and always focus on low-level temporal learning. But no explanations are given to prove this statement.
2. Comparing Eq.4 and Eq.8, the formats of these two equations are not unified. In addition, the meaning of the symbols used in the equations is not clearly given, making the equations hard to understand. Also, it is strange to use t (lower letter) to represent the features after the pooling operation while the upper letter T represents the time sequence. Similar problems occur in all the equation parts.
3. In equation 12, in my understanding, the m should equal Flatten(mβ)
4. The authors mention that the temporal dimension T is set to 16. However, no selection details are given. Suppose we have a 300 frames Deepfakes video with the last 150 frames tampered. The frames selection strategy would be crucial in detecting this Deepfakes video. In addition, no discussions for frame number selection are given. The authors should present more analysis on how frame number influences the proposed structure.
5. The following method is not compared in generalizability evaluation:
Kaede Shiohara, Toshihiko Yamasaki: Detecting Deepfakes with Self-Blended Images. CVPR 2022.

**Limitations:**

The authors have adequately addressed the limitations and potential negative societal impact of their work.

**Strengths And Weaknesses:**

Strengths:
The overall performance of the proposed method is outstanding.
Weaknesses:
The writing of the manuscript should be improved.

---

> ### Author Response · Authors · 2022-08-02
> **Author Response to Reviewer Yygr**
>
> We sincerely thank the reviewer for the constructive comments. We will respond to each detailed concern as follows:
>
> > **Q1: "In the discussion part for Section 3.2, the authors claim that the proposed ... can explicitly avoid semantic modeling of features like facial structure and always focus on low-level temporal learning. But no explanations are given to prove this statement."**
>
> **A1**:
>
> **1)** " explicitly avoid semantic modeling of features like facial structure". This statement is intuitively made. Since we split frames into independent squential patches, semantic information like facial structure is explicitly corrupted during modeling.
>
> **2)** "always focus on low-level temporal learning". This statement is related to the first one. We do not model the inter-patch relation (like original ViT) in the Local Sequence Transformer (LST), but encode only patch sequences separately.
>
>    a). Low-level: Considering that the facial semantics are explicitly excluded, our LST intuitively models low-level information. Moreover, we adopt shallow learnable 3D filters for further low-level information extraction instead of using deep convolutional layers which inevitably focus on high-level semantics.
>
>    b). Temporal: Moreover, temporal learning is achieved by self-attention operations working on the input of patch sequences.
>
> > **Q2: "Comparing Eq.4 and Eq.8, the formats of these two equations are not unified. ... it is strange to use t (lower letter) to represent the features after the pooling operation while the upper letter T represents the time sequence..."**
>
> **A2**: Thanks for pointing out the problems. We have revised these parts for better demonstration. Please find details in "General Responses" and the revised paper (marked in blue).
>
> **1)** The formats of  Eq.4 and Eq.8 were indeed described differently. We take the advice and revise the two equations to be unified for easier understanding.
>
> **2)**. In Eq (2), we used the subscript $pt$ of $\mathrm{x}_{pt}^{1,i}$ to distinguish from image patch $\mathrm{x}_{p}^{1,i}$. In the revision, we abandon the subscript and use a different symbol.
>
> **3)** In addition, we use the upper letter $T$ in superscript to describe dimension size, e.g., $m_o\in \mathbb{R}^{T\times H\times W}$, thus we also keep the same meaning as a constant of $T$ in all equations.
>
> **4)** " it is strange to use t (lower letter) to represent the features after the pooling operation". We have changed this notation in the revision.
>
> > **Q3: "In equation 12, in my understanding, the m should equal Flatten(mβ)"**
>
> **A3**: Thank the reviewer for pointing it out. We have fixed it in the revision.
>
> > **Q4: "The authors mention that the temporal dimension T is set to 16. However, no selection details are given ... The authors should present more analysis on how to frame number influences ..."**
>
> **A4**: We empirically select the clip length. On the one hand, too short a clip does not ensure enough temporal information, e.g., 4 consecutive frames almost identical to each other, thus leading to sub-optimal temporal learning. On the other hand, too long a clip is not necessarily better: 1) A longer clip costs more memory, and a smaller batch size also slow down the training convergence, or worse. 2) We found some of the videos in the test datasets contained scene switching. Considering the FPS of 24, a video clip in our experiments will not last 1 second, thus greatly avoiding the scene-switching problem. We include related ablation and discussion in the "General Responses" and the revised paper.
>
> > **Q5: "The following method is not compared in generalizability evaluation: Kaede Shiohara, Toshihiko Yamasaki: Detecting Deepfakes with Self-Blended Images. CVPR 2022."**
>
> **A5**: The mentioned CVPR22 paper [a] is indeed an encouraging concurrent work. However, it was not available on the CVPR22 website when we submitted our paper and was therefore missed. We have made comparisons with them by carefully runing their official code.
>
> | Method  | CelebDF | DFDC | FaceSh | DeepFo | Average |
> | ------- | ------- | ---- | ------ | ------ | ------- |
> | LTTD    | 89.3    | 80.4 | 99.5   | 98.5   | 91.9    |
> | SBI+EB4 | 89.9    | 74.9 | 97.4   | 77.7   | 85.0    |
>
> The generalization shown by both methods is excellent and our method outperforms SBI in DFDC, FaceSh, DeepFo with a clear margin. In addition, the contributions of the two papers are very different: SBI achieves generalization by creating training data (from the perspective of model training), while our approach focuses on learning more generalizable and robust features (from the perspective of feature learning). How to integrate the merits of both will be a direction worthy of further exploration in the future.
>
> [a] Detecting Deepfakes with Self-Blended Images. CVPR 2022

---

> > ### Comment · Reviewer_Yygr · 2022-08-09
> > **Keep My Original Score**
> >
> > The authors give more analysis about frame selection strategy. Also, the authors give the comparison between the proposed method and SBI. In addition, the equations are well written in the rebuttal version, my main concerns have been addressed.

---

### Official Review · Reviewer_zPSW · 2022-07-11

**Rating:** 5
**Confidence:** 4
**Soundness:** 3 good
**Presentation:** 3 good
**Contribution:** 3 good

**Summary:**

Briefly summarize the paper and its contributions. This is not the place to critique the paper; the authors should generally agree with a well-written summary.

To achieve generalizability across deepfake methods and robustness towards image post-processings, this paper proposes the Local- & Temporal-aware Transformer-based Deepfake Detection (LTTD) framework. The experiments show that the proposed approach achieves the state-of-the-art generalizability and robustness.

**Questions:**

See above.

**Limitations:**

This work aims to detect deepfake videos, which is important to prevent the spread of fake information.

**Strengths And Weaknesses:**

Strengths:
+ The paper attempts to ensure both generalizability and robustness in deepfake detection, which are two important problems in deepfake detection. The proposed method focuses on low-level temporal learning and prevents overfitting to global semantic cues.

+ The experiments are comprehensive.

+ The paper is clear to understand and the writing quality is relatively good.


Weaknesses:
- The paper claims that the method crops the face regions using the same bounding box randomly determine the clip range in-the-fly (L246), which may affect the practicality. It may be suitable for the videos in datasets because many of these videos often have a single scene. However, some videos in datasets and most videos in wild often have many scenes. If the clip contains scene switching, it may cause face misalignment and affect the detection results.

- Some details of selecting video clips are missing. How many clips extracted from one video? In FaceForensics++, one video may have less than 300 frames or more than 1000 frames. For short videos and long videos, if the number of clips different? The paper report video-level AUC Why the temporal dimension is 16? It is necessary to have an ablation study to see the effect of different choices of this parameter.

- The method splits the input image into local patches and use 3D convolutions. It seems that the framework needs complex computing. What about the efficiency of inference?

- It is not clear that how to compute the ground truth similarity matrix and how to generate the mask sequence (L201-205), especially the fake clip.

---

> ### Author Response · Authors · 2022-08-02
> **Author Response to Reviewer zPSW**
>
> We sincerely thank the reviewer for the constructive comments. We will respond to each detailed concern as follows:
>
> > **Q1: "The paper claims that the method crops the face regions using the same bounding box randomly determine the clip range in-the-fly, which may affect the practicality... If the clip contains scene switching, it may cause face misalignment and affect the detection results."**
>
> **A1**:
>
> **1)** Normally, scene or shot switching can be detected by off-the-shelf scene/shot detection tools.  Just like all faces are detected by face detectors in this task, we can also use scene/shot detection tools in the data preprocessing.
>
> **2)** We empirically set the clip length to 16 in this paper. Considering the FPS of 24, a video clip in our experiments will not last for 1 second. Such practice also ensures that most modeled clips are temporarily stable.
>
> > **Q2: "Some details of selecting video clips are missing. How many clips extracted from one video? ... Why the temporal dimension is 16? It is necessary to have an ablation study ..."**
>
> **A2**: The details are provided in the revision. Different from current SOTAs [56,57] using all the frames, for storage reasons, we extract only the first 128 frames of all videos in our experiments. Thus, the final prediction is an average of 8 clips. The clip length is emperically set to 16, please find the ablation in the "General Responses" and the revised paper.
>
> > **Q3: "... It seems that the framework needs complex computing. What about the efficiency of inference?"**
>
> **A3**: Please find the details about model complexity in the "General Responses". The proposed LTTD framework has a comparable model size and computing complexity with the SOTA methods. The main computation overhead of LTTD is arised from the dense linear connections of self-attention, in contrast, the shallow 3D convolutions contribute little.
>
> > **Q4: "It is not clear that how to compute the ground truth similarity matrix and how to generate the mask sequence (L201-205) ..."**
>
> **A4**:
>
> **1)** "how to generate the mask sequence". We denote the original mask sequence as  $m_o\in \mathbb{R}^{T\times H\times W}$, where the $T$ is the temporal dimension, i.e., there are $T$ masks, which are generated by simply subtracting the fake frame from the corresponding real one. We will add a short description with the "modification mask" we mentioned at L207 in the revision.
>
> **2)** The $sim_{gt}\in \mathbb{R}^{N\times N}$ (ground truth similarity matrix) is calculated from the mask sequence of corresponding video clip (Eq (12,13), where we fixed the bugs). Then, the  $sim_{gt}\in \mathbb{R}^{N\times N}$ is calculated by subtraction (Eq (13)), which measures the similarity of corresponding patches at different spatial regions. The value range of $sim_{gt}\in \mathbb{R}^{N\times N}$ is normalized to [-1,1], as the same as cosine similarity range (Eq (11)).

---

> > ### Comment · Reviewer_zPSW · 2022-08-09
> > **Keep original rating unchanged**
> >
> > After reading authors' feedback and other comments, I decide to keep my rating - borderline accept.
> >
> > Some of my concerns have been resolved that the authors provide more details, specially on the evaluation. Besides, the authors provide analysis on the model complex that there is no significant computation cost even though 3D convolutions are applied.
> >
> > Overall, this paper presents a new model for deepfake video detection and shows the state of the art results.

---

### Official Review · Reviewer_vJQN · 2022-07-11

**Rating:** 5
**Confidence:** 5
**Soundness:** 3 good
**Presentation:** 2 fair
**Contribution:** 3 good

**Summary:**

This paper proposes a Local- and Temporal-aware Transformer-based Deepfake Detection (LTTD) framework to capture temporal cues from local sequences. The authors design a Local Sequence Transformer (LST) models the temporal consistency of restricted spatial regions to learn local-to-global features. The proposed LTTD framework achieves outstanding generalizability and robustness.

**Questions:**

1. Novelty.
Many studies prove that the low-level patterns and temporal inconsistency are effective clues for Deepfake detection. What is the motivation of this search? Does the LTTD framework address specific problems that previous methods have neglected? The authors are supposed to underline the motivation and advantage of the proposed method.

2. Analyses.
1) The novel Low-level Enhancement is designed by shallow 3D convolutions. It can be regarded as an 3D extension of textural enhancement [60]. The authors should validate whether the 3D convolutions have the major contribution of improving feature learning. Can it be implemented with 2D convolutions to have comparable performance gain and less computational overhead?
2) The LTTD focuses on the low-level temporal patterns of restricted spatial region. What is the restricted spatial region? Please introduce the definition of restricted region. Furthermore, is it necessary to restrict specific regions? The authors should analyze the difference between restricted region and unrestricted region.
3) As shown in Table 3, the ViT [18] has inferior performance to the CNN baseline Xception. What is the reason for this phenomenon? How do the ViT-based methods FTCN [62] and LTTD improve the performance of ViTs on Deepfake detection? What is the key to this significant improvement?
4) Complexity. The comparisons on model complexity and GFLOPs between LipForensics, FTCN, and LTTD are desirable.
5) Localization. Can the LTTD localize the forged regions? It will be explicit to visualize the localization of face forgery.


**Limitations:**

Adequately claimed.

**Strengths And Weaknesses:**

Strengths:
1. Clear description of method.
2. Remarkable generalization towards unseen deepfakes and strong robustness against post-processing operations.

Weakness:
1. Novelty.
2. Analyses.

---

> ### Author Response · Authors · 2022-08-02
> **Author Response to Reviewer vJQN part(2/2)**
>
> > **Q4: "As shown in Table 3, the ViT [18] has inferior performance to the CNN baseline Xception. What is the reason for this phenomenon? How do the ViT-based methods FTCN [62] and LTTD improve the performance of ViTs on Deepfake detection?"**
>
> **A4**:
>
> **1)** "Why Xception outperforms ViT?". More precisely, Xception only demonstrate better performance on FF++ (in-dataset, AUC% 99.38 vs 97.92) and FaceSh (cross-dataset, AUC% 78.6 vs 65.56), while ViT shows better performance on more chanllenging DFDC (cross-dataset, AUC% 72.89 vs 67.36). On the one hand, Xception with more abundant inductive bias tends to better learn the specific forgery patterns or identity features in the train set, thus achieving better performance on FF++ and FaceSh (FaceSh shares the same identities with FF++, i.e., the same source videos are adopted to generate deepfakes). On the other hand, ViT with less inductive bias demonstrates better generalization, thus having advantages on DFDC evaluation. A similar discussion was made in Sec 4.5, where we drew the conclusion with a more intuitive visulization (Fig 3).
>
> **2)** "Why improvements of FTCN and LTTD compared with ViT?". First, although FTCN also employs a self-attention module, there are foundamental differences comparing with LTTD regarding both motivation and model design, in which the deep point-wise 3D convolution operations play a major role in FTCN. Moreover, we think the key to the significant generalization improvements of LTTD compared with ViT is the idea of `low-level & temporal feature learning` and the specially devised `learning-within-patch` model framework. For the idea of `low-level & temporal feature learning`, we have made a related response in the earlier comments (Novelty comment); for the effects of `learning-within-patch` framework, it can be learned from the ablation (Table 3), where the model w/o conv enhancement (LTTD w/o LST) already outperforms the two baselines. Moreover, Fig. 3  shows that our LTTD learns completely different features.
>
> > **Q5: "Complexity. The comparisons on model complexity and GFLOPs between LipForensics, FTCN, and LTTD are desirable."**
>
> **A5**: Please refer to the "General Responses".
>
> > **Q6: "Localization. Can the LTTD localize the forged regions? ..."**
>
> **A6**: Thanks for the advice. Our method can indeed localize the forged regions. We include a short discussion with visualizations in the revised supplementary material.
>
> **Follow-up**: We hope that our responses so far have cleared up the confusion for the reviewer to reevaluate our paper. We are willing to have further discussions if there is anything we could clarify.

---

> ### Author Response · Authors · 2022-08-02
> **Author Response to Reviewer vJQN part(1/2)**
>
> We sincerely thank the reviewer for the constructive comments. We will respond to each detailed concern as follows:
>
> > **Q1: "Novelty. Many studies prove that the low-level patterns ... What is the motivation of this search? Does ... address specific problems that previous methods have neglected? "**
>
> **A1**: We will restate our novelty and highlight our differences with previous studies here.
>
> **1)**  Previously, low-level patterns are studied [19,26,35,40] using hand-crafted low-level filters, which will be less effective on degraded data in the presence of commonly applied post-processing procedures like visual compression [20,34,57]. This suggests their **lack of *robustness*** (L33).
>
> In this paper, low-level patterns are extracted from a spatio-temporal view with fully learnable 3D filters. Moreover, our operations on local patches naturally avoid high-level semantic modeling. These designs  ***better adapt to the complex distributions of untapped deepfakes, making a more robust model*** (L43).
>
> **2)**  As for temporal inconsistency, many previous works [5,30,20,50,44] pursue to identify certain abnormal behaviors (e. g., abnormal eye blinking, phoneme-viseme mismatches, aberrant landmark fluctuation) (L26, L102). However, the remarkable visual forgery cues are expected to be gradually eliminated during the continuous army race between forgers and detectors. Considering the substantial temporal differences arise locally during the ***independent local modifications*** of forged frames, we propose to achieve deepfake detection by learning the local & low-level temporal inconsistency within a *restricted spatial space (16x16 patch)*. The proposed framework shows SOTA performance and good interpretability.
>
> **3)** A close discussion is presented in the Introduction section. In short, our way of modeling low-level and temporal information can address the dilemma between *robustness* and *generalization*. Our framework demonstrates promising performances and interpretable results (Fig. 3).
>
> > **Q2:  "The novel Low-level Enhancement ... can be regarded as an 3D extension of textural enhancement [60]. The authors should validate whether the 3D convolutions have the major contribution ... Can it be implemented with 2D convolutions ..."**
>
> **A2**:
>
> **1)** The introduced Low-level Enhancement was motivated by the previous low-level feature learning methods which use hand-crafted filters [40,19,26,35]. Our intuition here is to leverage shallow learnable filters for low-level information processing.
>
> **2)** On the other hand, different from [55] that uses stacks of deep convolutional layers,  we adopt only 1 layer of convolutional operation in each Low-level Enhancement Transformer stage. Thus only a few additional parameters are involved. Moreover, the performance gain is significant as shown in the Table 3, especially on the challenging DFDC dataset.
>
> **3)** The reason we use 3D filters instead of 2D is mainly related to the requirement of sequential modeling. Voxel-level alignment should be considered with cubic kernels. Here we replace the 3D filters with the 2D ones and show the results in the table below.
>
> | Kernel type$\downarrow$ Dataset $\rightarrow$ | FF++  | CelebDF | DFDC  | FaceSh | DeepFo |
> | --------------------------------------------- | ----- | ------- | ----- | ------ | ------ |
> | 2D                                            | 99.32 | 84.90   | 79.54 | 98.49  | 97.20  |
> | 3D                                            | 99.52 | 89.25   | 80.39 | 99.51  | 98.50  |
>
> It can be seen that 2D filters would lead to a inferior performance.
>
> > **Q3: "The LTTD focuses on the low-level temporal patterns of the restricted spatial region. What is the restricted spatial region? ..."**
>
> **A3**: As we split a video into squential patches, the "the restricted spatial region" refers to the area of space enclosed by the patch boundary. In this paper, we set the patch size to 16x16, which considerably corrupts the semantic information and thus is suitable for low-level pattern learning.

---

> ### Author Response · Authors · 2022-08-09
> **Looking forward to your reply**
>
> Hi reviewer,
>
> The discussion period is closing soon. Please take a look at our responses to your pre-rebuttal concerns. 1) Regarding novelty, we clarify the differences between this paper and related arts, where the key dilemma between **robustness** and **generalization** is resolved by the introduced low-level & temporal learning. 2) The asked analyses are also provided in the responses, revised paper, or the revised supp.
>
> We hope that our responses so far have cleared up the confusion for you to reevaluate our paper. We are willing to have further discussions until we can still reply.
>
> best

---

### Official Review · Reviewer_nqV7 · 2022-07-11

**Rating:** 6
**Confidence:** 4
**Soundness:** 2 fair
**Presentation:** 3 good
**Contribution:** 3 good

**Summary:**

The authors propose a framework to improve the generalization and robustness of deepfake detection, which relies on local low-level and temporal information, and transformer-based model. In particular, the Local Sequence Transformer (LST) is used to identify low-level temporal inconsistency and the Cross-Patch Inconsistency loss (CPI) is used to model spatial inconsistency. The Global Contrastive Classification (GCC) is used for final classification, which inserts temporal tokens and adopts additional three Transformer blocks.
Quantitative experiments show better results on four datasets in generalization evaluation benchmark and on seven perturbations in robustness evaluation benchmark, compared with some recent works.


**Questions:**

Confusion:
1. In Sec 3.3, it seems that the supervision information has not only binary labels but also masks, so the prediction may be determined by other simpler structures like fine-grained classification layers, instead of the proposed complex “temporal tokens + three Transformer blocks” structure.
2. Based on Table 3, the result line from the ‘LTTD w/o CPI’ shows that CPI contributes far less than the LST or GCC module in DFDC and DeepFo. So is the CPI module significance?

Limitation:
1. For the authors’ motivation, using low-level features instead of semantic features benefits a lot in Deepfake Detection. It seems that the middle-level features, and the suitable mixture of low-& middle-level features may also contribute to Deepfake Detection. It is a pity that the paper does not include this exploration.
2. For Sec 4.4, the robustness evaluation seems not strong enough, the authors could refer to the benchmarks in Improving [45] and ForgeryNet [CVPR 2021, Forgerynet: A versatile benchmark for comprehensive forgery analysis]，whose robustness evaluation contains more types of perturbations and mix perturbation test sets.


**Limitations:**

The authors have done it properly.

**Strengths And Weaknesses:**

Strengths:
1. The paper is well written, easy to follow and provides adequate experimental results.
2. The proposed framework successfully integrates temporal information and low-level features, which are separate from improved vision transformer and shallow CNN, and could be of some contributions to the community.
3. There are some novel ideas in the LST module: the first is the input type: ‘local patches with same spatial position’, and the second is the low-level temporal enhancement module, corresponding with the research that the low-level artifact is more suitable than the semantic artifact for deepfake detection. What’s more, the CPI module is well designed, and the motivation of the GCC module shows good understanding of deepfake datasets.
4. The evaluation experiments of generalization and robustness are well done.

Weakness：
1. In Sec 3.2, the authors use ‘shallow 3D convolution’ in the LST module for ‘align’ as the first reason (Ignoring the second reason temporally). In my opinion, it is similar to face alignment, so if face-alignment operation plays the same role，why not just do it in the pre-processing stage.
2. In Sec 3.3, sim_gt is calculated by interpolation operation, how about max or average pooling operation? The authors may provide ablation study about how to calculate it.
3. In Sec 4, the experiment part is lacking intra-evaluation, for example, both training and testing on FF++.
4. In Sec 4.1, how many frames or clips are sampled from videos in each deepfake dataset.
5. In Sec 4.3, the result analysis about robustness evaluation seems not enough.
6. In Sec 4.4, the ablation study is only done for generalization evaluation, that for robustness evaluation is also required.

---

> ### Author Response · Authors · 2022-08-02
> **Author Response to Reviewer nqV7 part(2/2)**
>
> > **Q6: "the ablation study for robustness evaluation is also required."**
>
> **A6**:
>
> | Method       | Clean | Color Saturation | Color Contrast | Block-Wise Noise | Gaussian Noise | Gaussian Blur | Pixelation | Video Compression | Avg/Drop   |
> | ------------ | ----- | ---------------- | -------------- | ---------------- | -------------- | ------------- | ---------- | ----------------- | ---------- |
> | LTTD         | 99.4  | 98.9             | 96.4           | 96.1             | 82.6           | 97.5          | 98.6       | 95.0              | 95.0/-4.3  |
> | Face X-ray   | 99.8  | 97.6             | 88.5           | 99.1             | 49.8           | 63.8          | 88.6       | 55.2              | 77.5/-22.3 |
> | LTTD w/o LST | 98.8  | 93.9             | 92.6           | 86.0             | 68.8           | 93.2          | 95.9       | 91.7              | 88.9/-9.9  |
> | LTTD w/o CPI | 98.8  | 93.9             | 92.6           | 86.2             | 68.9           | 93.2          | 95.1       | 91.7              | 88.8/-10.0 |
> | LTTD w/o GCC | 99.1  | 97.6             | 90.6           | 94.9             | 76.0           | 89.0          | 97.4       | 91.4              | 90.9/-8.2  |
>
> From the results, we find that the special designs all contribute to optimal performance. For color contrast (CC), the Global Contrastive Classification (GCC) module has a more significant contribution as it better enhances the detection of local color anomalies by modeling features in different spatial regions through global comparisons. In contrast to block-wise noise (BW), the Local Sequence Transformer (LST) and Cross-Patch Inconsistency (CPI) modules contribute more, since BW noise affects only a very small local area, it has no effect on the low-level & temporal features in other regions. However, it will interfere with the global contrast learning of GCC. The results on gaussian noise (GNC) can be understood consistently with BW. Since GNC comprehensively modifies the low-level features,  low-level & temporal learning of LST and CPI will be greatly affected, while the global contrastive learning of GCC is less affected, thus leading to a more significant contribution of GCC. Compared with Face X-ray focusing on spatial low-level feature learning, the performance degradations of our models are significantly smaller due to the consideration of temporal dimension. This phenomenon is also in line with our motivation we discussed in the Sec. Introduction that low-level features are susceptible to perturbations and robustness will be enhanced by incorporating temporal learning. We have added this part to the revised supplementary material.
>
> > **Q7: "It seems that the supervision information has not only binary labels but also masks, so the prediction may be determined by other simpler structures like fine-grained classification layers?"**
>
> **A7**: In Sec 3.3, we introduce CPI loss using the *modification masks* (which are created by simply subtracting the fake frame from the corresponding real one) as supervisions, but they do not directly affect the final prediction and are not be used while testing. CPI loss is only calculated for training regularization. The final prediction is given by the GCC module as described in L214.
>
> > **Q8: "Based on Table 3, the result line from the ‘LTTD w/o CPI’ shows that CPI contributes far less than the LST or GCC module in DFDC and DeepFo."**
>
> **A8**: LST and GGC modules are indeed the key components of our LTTD considering the main idea of low-level & temporal learning and local-to-global prediction. CPI provides only auxiliary contrastive supervision, which is helpful but relatively less significant.
>
> > **Q9："For the authors’ motivation, using low-level... It seems that the middle-level features, and the suitable mixture of low-& middle-level features may also contribute to Deepfake Detection."**
>
> **A9**: Our current version focuses specifically only on the low-level part. However, we think the idea of "suitable mixture of low-& middle-level features" is interesting and would be of great value.  We will add this to our discussions on future work directions.
>
> > **Q10: "For Sec 4.4, ... the authors could refer to the benchmarks in Improving [45] and ForgeryNet ..."**
>
> **A10**: Thanks for the advice. With limited time, we are unable to provide an evaluation with [a] and [b]. We would include a discussion with these benchmarks in the future.
>
> [a] Improving the efficiency and robustness of deepfakes detection through precise geometric features.
>
> [b] Forgerynet: A versatile benchmark for comprehensive forgery analysis
>
> Please do not hesitate to let us know if there are any additional clarifications or experiments that we can offer!

---

> > ### Comment · Reviewer_nqV7 · 2022-08-08
> > **Keep my score unchanged**
> >
> > Thanks for the response. I appreciate the authors for the effort and I believe that the results and discussion above can improve the quality of the submission.
> >
> > My major concerns are mostly addressed (on ‘mask’ and ‘CPI’), but as the other reviewers point that,
> > 1. Novelty: Not so significant.
> > 2. Robustness (my Q10): The performance of the model, when facing wild videos, in particular mixed degradation ones, which are closer to the real-world deepfake detection scene, is not sure.
> >
> >
> > To sum up, I keep my original score.

---

> ### Author Response · Authors · 2022-08-02
> **Author Response to Reviewer nqV7 part(1/2)**
>
> We sincerely thank the reviewer for the constructive comments. We will respond to each detailed concern as follows:
>
> > **Q1:  "In Sec 3.2, the authors use ‘shallow 3D convolution’ in the LST module for ‘align’... In my opinion, it is similar to face alignment, ... why not just do it in the pre-processing stage."**
>
> **A1**: We intentionally did not align faces because alignment errors are inevitable during per-frame processing. As a result, per-frame face alignment will certainly corrupt the natural low-level temporal consistency/inconsistency of both *real* and *fake* videos.
>
> > **Q2: "In Sec 3.3, sim_gt is calculated by interpolation operation, how about max or average pooling operation?"**
>
> **A2**: Interpolation is employed here only for narrowing the spatial dimensions of $\mathrm{sim}_{gt}$. We have tried pooling and found virtually no difference.
>
> >**Q3: "In Sec 4, the experiment part is lacking intra-evaluation, both training and testing on FF++."**
>
> **A3**: In recent works, in-dataset results are almost saturated (overfitting to specific kinds of artifacts may lead to better in-dataset performance, but worse generalization) and are not listed for comparison as a common practice. Our method achieves 99.52 AUC% on FF++, which also demonstrates SOTA in-dataset performance.
>
> > **Q4: "In Sec 4.1, how many frames or clips are sampled from videos in each deepfake dataset."**
>
> **A4**: For storage reasons, we extract only the first 128 frames of all videos in our experiments. Therefore, the final prediction is averaged from 8 clips. We have added these details in the revision.
>
> > **Q5: "In Sec 4.3, the result analysis about robustness evaluation seems not enough."**
>
> **A5**: Regarding robustness, we focus our analysis on our comparisons with Face X-ray and LipForensics, as both these studies are closely related to our discussions in the Introduction (about the dilemma of simultaneously achieving generalization and robustness).
>
> **1)** Low-level feature learning could lead to better generalization, but worse robustness. Face X-ray is one of the first works to achieve generalizable deepfake detection, focusing on low-level (blending boundary) learning. From the results in Table 2, Face X-ray suffers from drastic performance degradation when perturbations of gaussian noise, gaussian blur, and video compression are applied. The reason is that the low-level features of blended boundaries are greatly corrupted by these perturbations, thus clearly demonstrating the weakness of low-level feature learning in terms of robustness.
>
> **2)** Semantic feature learning will result in better robustness. Since most perturbations do not change the semantic information. LipForensics, which focuses on high-level understanding, demonstrates better robustness compared to Face X-ray under perturbations like gaussian noise, gaussian blur, and video compression. However, this method cannot generalize to scenes when the mouth part is blocked or even shut down.
>
> **3)** As discussed in the *Introduction* section, to achieve both *generalization* and *robustness*, we combine the learning of low-level & temporal features. The learned local patterns of our model are less influenced by low-level perturbations compared to Face X-ray. Moreover, our method performs better than LipForensics under different scenes.

---

### Author Response · Authors · 2022-08-02
**General Responses**

We thank all reviewers for the detailed and constructive comments. Here we first respond to some common questions.

Note: all the referred equations, lines, and citations in the responses correspond to the revised paper. Although we retain related descriptions of the reviewers in the quoted *Questions*.

## Notations and Equations

As advised by the reviewers, we revise some of the notations for better demonstrations. The paper is modified from four aspects:

1. Simplify the subscripts, which are dispensable, e.g., $p$ in $\mathrm{x}_p^{t,i}$ is removed as $\mathrm{x}^{t,i}$.
2. Change the symbol to avoid confusing subscripts, e.g., we change $\mathrm{x}_{pt}^{t,i}$ to $y^{t,i}$ to denote the patch features after low-level & temporal enhancement.
3. Unify the two sets of equations (Eq (1-9) in the revision).
4. Fix bugs, e.g., Eq(12,13) in the revision.

These modifications are marked in **blue**, please find the details in the revised paper.

Meanwhile, we notice what the reviewer said about the complex superscripts. In order to accurately describe both temporal and spatial dimensions, we have to use two superscripts, e.g., $\mathrm{x}^{t,i}$ denotes the image patch at  $i$-th spatial region of the $t$-th frame. But this is not really difficult to understand, since we use $t$ for a timestamp and $i$ for a spatial location consistently throughout the paper. We also include a short description in the revised version of Sec 3.1 Problem statement.

## Clip Length & Sampling Space Ablation

As suggested by the reviewers, we further conduct ablations on the hyper-parameters regarding the model input. In the pre-rebuttal version, the two hyper-parameters (clip length, frame sampling space) of the input size are set empirically. On the one hand, too short a clip does not ensure enough temporal information, e.g., 4 consecutive frames are almost identical to each other. On the other hand, too long a clip is not necessarily better: 1) A longer clip costs more memory, and a smaller batch size also slows down the training convergence, or worse. 2) We found some of the videos in the test datasets contained scene switching. Considering the FPS of 24, a video clip in our experiments will not last 1 second, thus greatly avoiding the scene-switching problem. Two hyper-parameters are further investigated: clip length and frame sampling space.

| Clip Length | Sampling Space | FF++  | DFDC  | DeepFo | Avg   |
| ----------- | -------------- | ----- | ----- | ------ | ----- |
| 8           | 1              | 99.26 | 77.25 | 96.82  | 91.11 |
| 16          | 1              | 99.52 | 80.39 | 98.50  | 92.80 |
| 32          | 1              | 99.47 | 80.92 | 98.41  | 92.93 |
| 64          | 1              | 99.38 | 79.23 | 98.49  | 92.37 |
| 16          | 2              | 99.15 | 76.17 | 97.32  | 90.88 |
| 16          | 4              | 98.51 | 73.02 | 91.33  | 87.62 |

From the results, the first 4 rows show that clip length has little effect on the results. We believe this is closely related to the idea of low-level temporal learning, which does not require the video clip to last long enough (e. g., 3 seconds) but only adequate frames to extract the low-level temporal patterns. Sparse sampling is another practice to aggregate more content in a single clip. When we expand the frame interval, the performance degrades considerably. This phenomenon suggests that more sparse sampling is detrimental to the learning of low-level temporal patterns even when more motion content is included. These findings demonstrate that LTTD is distinctly different from related temporal-based models considering the low-level temporal learning is specially designed for deepfake detection.

## Complexity Analysis

| Method              | LipForensics | FTCN  | LTTD w/o Low-enhancement | LTTD  |
| ------------------- | ------------ | ----- | ------------------------ | ----- |
| Num. of Params. (M) | 36.00        | 26.47 | 20.57                    | 22.66 |
| GFLOPs              | -            | 8.25  | 13.25                    | 14.51 |

We show the model complexities of two SOTA models, and two of our models in the table (the num. of params. of LipForensics is cited from [57]).

1. Comparing LTTD w/o Low-enhancement and LTTD, the increased parameters and FLOPs correspond primarily to the linear layers of the introduced Low-level Enhancement Transformer stages.
2. Compared with the SOTA models, our LTTD achieves the best performance (Table 1 in the paper) with a minimal number of parameters. The FLOPs of our model is relatively higher considering the intensive computation of the linear layers we use.
3. In short, our model reports better performance with a smaller number of parameters, although the inference may be slower.

---

### Author Response · Authors · 2022-08-07
**Further questions from the reviewers?**

Dear reviewers,

Thanks again for your efforts and valuable comments. We have provided corresponding responses and results, which we hope have covered your concerns. Since the discussion period is closing, we would like to know if you have any lingering concerns? If there is more we could do to help you make your final decision, please let us know.

If you are satisfied with our responses, please consider raising your scores and let our ideas contribute to the development of deepfake detection.

best

---

### Meta-Review · Area_Chair_pwnV · 2022-08-25

**Recommendation:** Accept
**Confidence:** Certain

**Metareview:**

All reviewers are positive about this paper. Generally speaking, the proposed method is novel and is also easy to follow due to well writing. Also, the experiments are comprehensive. In the rebuttal, the authors also provide some qualitative results to clearly respond to the concerns of reviewers. So, I suggest accepting this paper.

**Award:**

No

---

### Decision · Program_Chairs · 2022-09-14

Accept